# A high-resolution morphological and ultrastructural map of anterior sensory cilia and glia in *Caenorhabditis elegans*

David B Doroquez[1,2][†], Cristina Berciu[1,3][†], James R Anderson[4], Piali Sengupta[1,2]*, Daniela Nicastro[1,3]*

[1]Department of Biology, Brandeis University, Waltham, United States; [2]National Center for Behavioral Genomics, Brandeis University, Waltham, United States; [3]Rosenstiel Basic Medical Sciences Research Center, Brandeis University, Waltham, United States; [4]Department of Ophthalmology, John A Moran Eye Center, University of Utah School of Medicine, Salt Lake City, United States

**Abstract** Many primary sensory cilia exhibit unique architectures that are critical for transduction of specific sensory stimuli. Although basic ciliogenic mechanisms are well described, how complex ciliary structures are generated remains unclear. Seminal work performed several decades ago provided an initial but incomplete description of diverse sensory cilia morphologies in *C. elegans*. To begin to explore the mechanisms that generate these remarkably complex structures, we have taken advantage of advances in electron microscopy and tomography, and reconstructed three-dimensional structures of fifty of sixty sensory cilia in the *C. elegans* adult hermaphrodite at high resolution. We characterize novel axonemal microtubule organization patterns, clarify structural features at the ciliary base, describe new aspects of cilia–glia interactions, and identify structures suggesting novel mechanisms of ciliary protein trafficking. This complete ultrastructural description of diverse cilia in *C. elegans* provides the foundation for investigations into underlying ciliogenic pathways, as well as contributions of defined ciliary structures to specific neuronal functions.

**\*For correspondence:**
sengupta@brandeis.edu (PS);
nicastro@brandeis.edu (DN)

[†]These authors contributed
equally to this work

**Competing interests:** The
authors declare that no
competing interests exist.

**Reviewing editor**: Oliver Hobert,
Columbia University, United
States

## Introduction

Animals must sense and respond to multiple environmental cues over a wide range of signal intensities. The complexity of external cues is reflected in part in the remarkable diversity of sensory neuron morphologies and functions. Many major sensory neuron types contain microtubule (MT)-based primary cilia that house signal transduction molecules and are essential for the neurons' sensory properties (*Perkins et al., 1986*; *Inglis et al., 2007*; *McEwen et al., 2008*; *Ramamurthy and Cayouette, 2009*; *Pifferi et al., 2010*). Although basic assembly mechanisms, core structures, and overall functions of cilia are highly conserved, sensory neuron cilia are structurally diverse and specialized for their unique roles (*Berbari et al., 2009*; *Silverman and Leroux, 2009*). For example, vertebrate olfactory neurons contain a tuft of 6–17 cilia that emanate from the dendritic knobs and house olfactory signaling proteins (*Menco, 1980*; *McEwen et al., 2008*; *Pifferi et al., 2010*). Similarly, rod and cone photo-receptors contain highly elaborate ciliary outer segments that differ in morphology, and that localize molecules required for phototransduction (*Berbari et al., 2009*; *Yildiz and Khanna, 2012*). Sensory cilia morphology and function are further regulated via interactions with non-neuronal cells such as glia (*Young and Bok, 1969*; *Strauss, 2005*; *Bacaj et al., 2008*; *Procko et al., 2011*). Defects in cilia structure and function lead to altered cellular signaling and contribute to systemic disorders collectively called ciliopathies (*Green and Mykytyn, 2010*; *Louvi and Grove, 2011*; *Waters and Beales, 2011*). Thus, analysis of sensory cilia architecture, as well as their organization and interaction with supporting

**eLife digest** To survive, animals must constantly gather information about their surroundings and then decide how to respond. Animals rely on cells called sensory neurons to help them perceive and process this information, and these neurons in most animals have smaller structures called cilia that help them to gather this information. The structures of these cilia can range from simple hair-like rods to complex branched arbors. Defective cilia can lead to cell degeneration and death.

Scientists have identified and determined the functions of many of the 60 sensory neurons with cilia in *C. elegans*, a tiny roundworm with a simple nervous system. These experiments have revealed that the shapes of these cilia are quite diverse, and that the shape determines the type of information the neurons process. Learning more about how cilia are shaped, and how these shapes allow them to perform specific sensory functions, would give scientists a better understanding of how the brain processes sensory information.

Doroquez et al. have now taken advantage of advances in imaging technology to generate highly detailed three-dimensional reconstructions of the cilia on 50 neurons in the nose of *C. elegans*. The experiments involved rapidly freezing the worms, slowly replacing the frozen water molecules with a preservative solution, and then embedding in resin. This allowed Doroquez et al. to slice the samples into very thin sections—some 1400 times thinner than a sheet of paper—and then image them with transmission electron microscopy and electron tomography. Finally, all these images were combined in a computer to produce 3D models of the cilia.

The models reveal a wide range of cilia structures, including some that have never been examined in detail before. Doroquez et al. were also able to see detailed structures within the cilia, including compartments that determine which proteins should enter into, or be excluded from, an individual cilium. The models, along with the results of previous studies, suggest that cilia are shaped by genetic factors and also by interactions with the environment. This detailed description of diverse cilia structures should now allow researchers to identify the genes that determine their unique shapes, and explore how specific shapes contribute to specific sensory functions.

cells, is essential for a complete description and understanding of the mechanisms that underlie functional specialization in the sensory nervous system.

The nematode *Caenorhabditis elegans* is an excellent model system in which to explore the molecular and neuronal mechanisms by which animals detect and respond to environmental cues (e.g., *Bargmann, 2006*; *Goodman, 2006*; *Zhang, 2008*; *Garrity et al., 2010*). The *C. elegans* hermaphrodite has a compact sensory nervous system comprised of at least 78 neurons of 34 types that differ in their morphologies, connectivities, and sensory properties (*Ward et al., 1975*; *Ware et al., 1975*; *Perkins et al., 1986*; *White et al., 1986*) (www.wormatlas.org). Many of the major chemo-, mechano- and thermosensory neurons are present in sensory organs located in the head and tail of the worm; 60 of these sensory neurons are ciliated (*Ward et al., 1975*; *Ware et al., 1975*; *Perkins et al., 1986*; *White et al., 1986*; *Bargmann, 2006*; *Inglis et al., 2007*) (www.wormatlas. org). As in other organisms, development, and maintenance of correct cilia structure is essential for the unique sensory properties of each neuron type in *C. elegans* (*Dusenbery et al., 1975*; *Lewis and Hodgkin, 1977*; *Culotti and Russell, 1978*; *Perkins et al., 1986*). In the head, ciliated sensory neurons are organized in several sensilla, each of which contains different sensory neuron types and associated glial support cells (*Ward et al., 1975*; *Ware et al., 1975*; *Perkins et al., 1986*). Specific functions have been assigned to many sensory neurons (*Bargmann, 2006*) (www.wormatlas.org), providing an experimentally amenable system in which to not only correlate structural and functional specialization, but to also describe in detail the extent of diversity in sensory cilia morphologies and ultrastructures present in a metazoan.

Pioneering work in the 1970s and 1980s provided an initial morphological and ultrastructural description of the anterior sensory anatomy of *C. elegans* (*Ward et al., 1975*; *Ware et al., 1975*; *Perkins et al., 1986*). In these studies, serial section transmission electron microscopy (ssTEM) of chemically fixed animals was used to describe the structure and organization of head sense organs. In particular,

these studies described the morphologically diverse cilia present at the dendritic endings of sensory neuron types in these sensilla (*Ward et al., 1975*; *Ware et al., 1975*; *Perkins et al., 1986*). For example, eight amphid organ sensory neurons were shown to contain structurally simple cilia that are directly exposed to the environment through a 'channel' formed by glial cells, whereas the cilia of other amphid sensory neurons were shown to be more complex and embedded within glial cell processes (*Ward et al., 1975*; *Ware et al., 1975*; *Perkins et al., 1986*).

Although these studies provided the basis for extensive further investigations into the sensory biology of *C. elegans*, two issues prompted us to revisit the description of the wild-type anterior sensory anatomy. First, advances in imaging and tissue preservation techniques have led to the development of methods and instrumentation for ultrastructural imaging that provide far more detailed and higher resolution views than was previously possible (*Rostaing et al., 2004*; *Weimer, 2006*; *Muller-Reichert et al., 2008*; *Hall et al., 2012*). As an example, serial section electron tomography (ssET) now allows for three-dimensional (3D) reconstruction of structures, and reveals novel features, due to almost isotropic resolution within the reconstruction rather than being limited in the *z*-direction by the thickness of plastic sections (*McEwen and Marko, 2001*; *Muller-Reichert et al., 2010*). Second, the ultrastructural features of several sensory cilia have remained poorly described, in part due to limitations in the visualization methods, but also due to issues arising from conventional chemical fixation of biological structures. For instance, the morphology and ultrastructure of cilia in the BAG gas-sensing and FLP nociceptive neurons are poorly described, and the axonemal MT distribution pattern in highly branched cilia, such as those in the AWA chemosensory neurons, have not been fully characterized.

Here, we provide a high-resolution morphological and ultrastructural analysis of the anterior endings of sensory neurons and glia in the adult *C. elegans* hermaphrodite using ssTEM and ssET of high-pressure frozen and freeze-substituted (HPF-FS) adult animals. This analysis has now allowed us to robustly model the morphologies of all anterior sensory cilia, comprising 50 of 60 ciliated neurons in the *C. elegans* adult hermaphrodite. We describe the ultrastructures of diverse cilia types in unprecedented detail including those of previously uncharacterized cilia, and describe new features of cilia–glial interactions. We also characterize their intercellular relationships, and identify important ultrastructural features including MTs, fibers, vesicles, and junctions. Moreover, this high-resolution map reveals three distinct and novel MT distribution processes in branched cilia, suggesting diverse modes of ciliary protein transport and delivery. Together with the extensive analyses of cell specification pathways possible in *C. elegans*, this complete ultrastructural description of the anterior sensory anatomy will allow for a detailed understanding of the molecular and cellular mechanisms by which morphological diversity is established, as well as the contribution of this diversity to cellular functional specialization.

## Results and discussion

### 3D reconstruction of the anterior sensory anatomy

HPF-FS has been shown to significantly reduce fixation artifacts and morphological deterioration of *C. elegans* samples for EM as compared to conventional chemical fixation techniques (*Muller-Reichert et al., 2003*, *2008*; *Weimer, 2006*). Consistent with these observations, we obtained excellent preservation of tissues from adult *C. elegans* hermaphrodites using HPF-FS (*Figure 1A*, *Figure 1—figure supplement 1*). In particular, we noted that: (a) subcellular structures in muscle were maintained, allowing resolution of the paracrystalline organization of actin-myosin fibers in sarcomeres (*Figure 1—figure supplement 1A*); (b) cell–cell connections as in the electron-dense *C. elegans* apical junctions (CeAJs) (*Labouesse, 2006*; *Lynch and Hardin, 2009*) or hemi-desmosome-like junctions were preserved with clearly visible membranes and cytoskeletal filaments (*Figure 1—figure supplement 1B,C*) (*Cox and Hardin, 2004*); (c) ciliary membranes exhibited well-defined, smooth structures with clearly resolved bilayers as compared to scalloped membranes observed in chemically fixed samples (*Figure 1—figure supplement 1D*); (d) doublet and singlet MTs could be observed, allowing identification of different protofilament (pf) numbers and inner MT structures (*Figure 1—figure supplement 1D*); and (e) MT-associated fibers (Y-links) could be identified in the ciliary transition zone (TZ), a morphologically distinct structure at the base of cilia that acts as a ciliary diffusion barrier (*Figure 1—figure supplement 1E*) (*Reiter et al., 2012*; *Szymanska and Johnson, 2012*). We did not observe obvious cell shrinkage or ice damage, which would be indicated by empty spaces

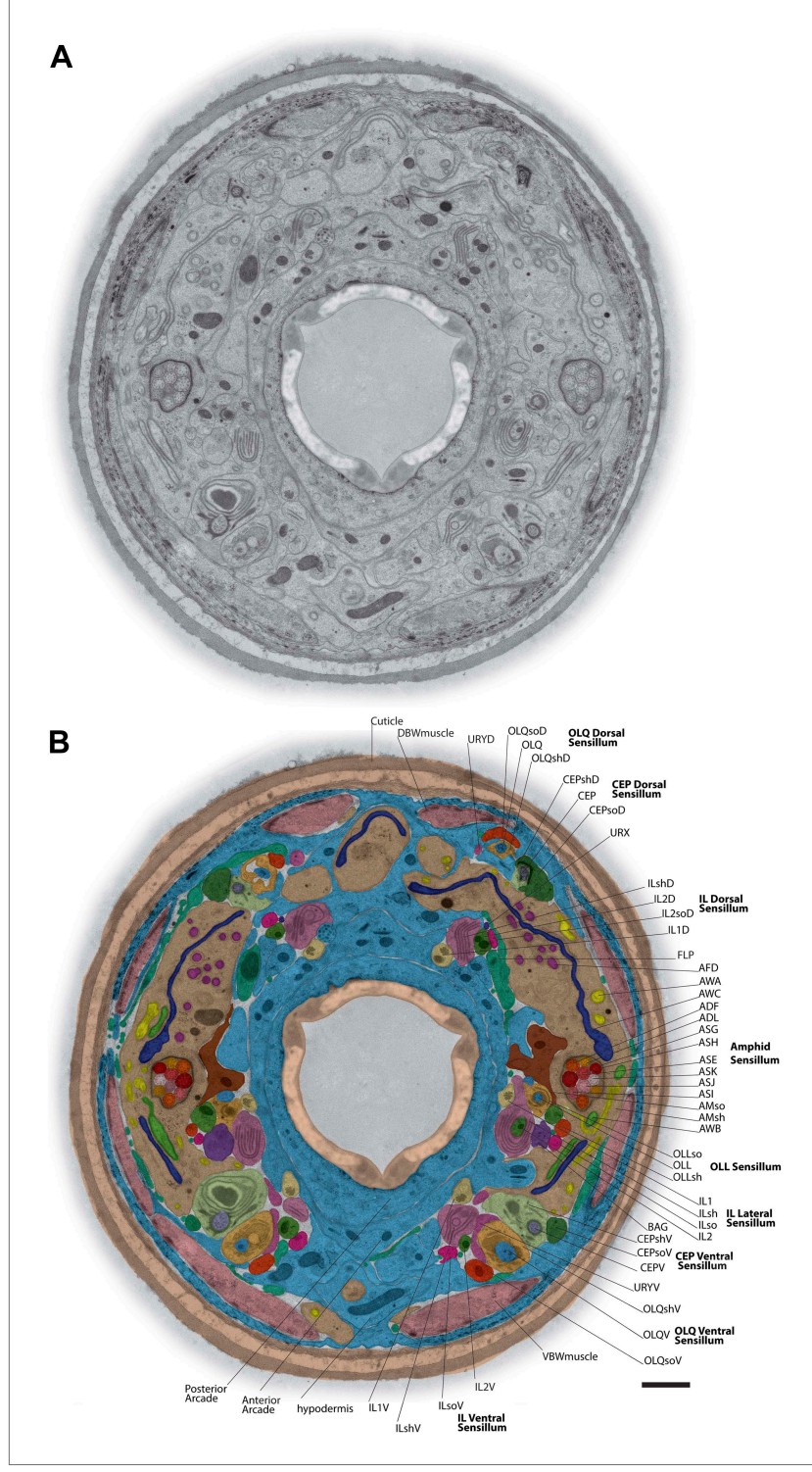

**Figure 1**. TEM cross-section with identified ciliary and glial endings. (**A**) Example TEM cross-section of a high-pressure frozen/freeze-substituted (HPF-FS) *C. elegans* hermaphrodite animal (70-nm cross-section, 5.9 μm from the anterior nose tip). Edges surrounding the cuticle were feather-cropped. See ***Figure 1—figure supplement 1*** for examples of preservation of subcellular structures in cilia, muscle, and junctions. (**B**) Endings of identified cells in the indicated bilateral sensilla within the cross-section TEM; only the right side of the animal is labeled. Cell endings are marked with a false-color overlay. Dorsal up, ventral down. Scale bar: 1 μm. See ***Figure 1—figure supplement 2*** for additional cross-section views.

*Figure 1. Continued on next page*

*Figure 1. Continued*

The following figure supplements are available for figure 1:

**Figure supplement 1**. Ultrastructural features maintained by HPF-FS.

**Figure supplement 2**. Example TEM cross-section images of the *C. elegans* nose.

surrounding cells, or the destruction or aggregation of filamentous structures such as those of muscle and cell junctions, respectively.

We reconstructed the 3D structure of the distal head region (~14 μm from the nose tip), containing the sensory endings of many sensory neuron types and glia, for three serially sectioned wild-type adult worms that were grown and prepared under identical conditions. We also imaged selected sections of the head region for five additional wild-type animals in cross-section (*Figure 1—figure supplement 2*), and one animal in longitudinal section. For each ssTEM 3D reconstruction, ~170 cross-sections of 70-nm thickness were collected on grids, while preserving the section sequence from proximal to distal. To fit an entire cross-section into a single EM image would require a relatively low EM magnification, which provides a large field of view but sacrifices image resolution. To instead take advantage of the excellent sample preservation by HPF-FS and thus the inherent increased resolution, we digitally recorded a montage of >100 EM images for each section at high magnification, providing a pixel size well below the resolution limit set by the sample preparation. The individual image tiles of the montage of each section were aligned and stitched together to generate a single 2D cross-section image (with a size of up to 20K × 20K pixels); images were then sequentially aligned to generate an ssTEM 3D reconstruction (see 'Materials and methods' for additional details). We then followed and outlined the cell membranes of the anterior processes of more than 90 cells, including those of sensory neurons and associated glial cells, on these serially aligned cross-sections to generate a complete 3D graphical model of the cellular organization of the *C. elegans* anterior anatomy (*Figure 2*, *Figure 3A,A'*, *Video 1*).

## Overall architecture of the anterior sensory anatomy

### Organization of neuronal endings in anterior sensilla

The dendrites of many ciliated and non-ciliated sensory neurons terminate distally in a stereotypical pattern at two or more of the six symmetrical dorsal, ventral, and lateral lips around the roughly triangular opening of the worm mouth (*Ward et al., 1975*; *Ware et al., 1975*; *Perkins et al., 1986*) (*Figure 1*, *Figure 2*, *Figure 1—figure supplement 2*). Consistent with previous reports (*Ward et al., 1975*; *Ware et al., 1975*; *Perkins et al., 1986*), we observed stereotypical positioning of cellular anterior processes across worms when comparing sections from similar regions (distal region: compare *Figure 1—figure supplement 2C,D*, middle region: compare *Figure 1B*, *Figure 1—figure supplement 2B*, proximal region: compare *Figure 1—figure supplement 2E,F*).

The largest sensilla in the worm nose are the bilateral amphid sensory organs, each of which contains 12 ciliated sensory neurons (*Ward et al., 1975*; *Ware et al., 1975*; *Perkins et al., 1986*). Although the dendrites of these amphid neurons are bundled for most of their length in the lateral regions of the worm, the cilia of some amphid neurons spread out considerably and project into the dorsal and/or ventral lips (*Figure 1*, *Figure 2*, *Figure 1—figure supplement 2*). In addition to the lateral amphid openings, each of the six lips that surround the mouth opening contains two labial sensory organs: (a) an inner labial (IL) sensillum that terminates close to the mouth (inner lip) and contains two ciliated neurons, IL1 and IL2 (*Figure 1*, *Figure 2*, *Figure 1—figure supplement 2*); and (b) an outer labial sensillum whose pore is situated toward the outside of the lips (*Ward et al., 1975*; *Ware et al., 1975*; *Perkins et al., 1986*). The lateral outer labial (OLL) sensilla each contain one ciliated neuron OLL, whereas each of the dorsal and ventral outer labial quadrant (OLQ) sensilla contain one ciliated OLQ neuron (*Figure 1*, *Figure 1—figure supplement 2*) (*Ward et al., 1975*; *Ware et al., 1975*; *Perkins et al., 1986*). Each OLQ sensillum is additionally paired with a cephalic (CEP) sensillum, which includes one ciliated CEP neuron (*Figure 1*, *Figure 1—figure supplement 2*) (*Ward et al., 1975*; *Ware et al., 1975*; *Perkins et al., 1986*). Although the neurons of the OLQ and CEP sensilla are in close proximity, they are considered to be separate sensory organs, because they are associated with independent sets of glial support cells, as compared to the two IL neurons that share one set of support cells (*Figure 3—figure supplement 1*).

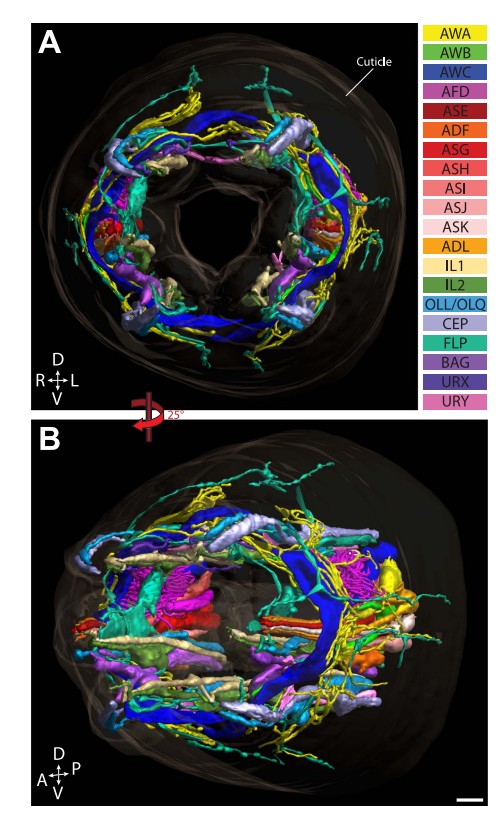

**Figure 2**. Graphical model of a high-resolution ssTEM 3D reconstruction of the anterior sensory endings. 3D reconstruction of the cilia and dendritic endings of anterior sensory neurons modeled from 166 thin serial sections (14 µm, starting at the nose). Front (**A**) and angled (**B**) profile views are shown. Cell projections are color-coded as indicated in upper right panel. D, dorsal; V, ventral; A, anterior; P, posterior. Scale bar: 1 µm.

Two additional, bilateral ciliated neurons, BAG and FLP, share the unique feature that they are closely associated with the support cells of the lateral IL sensilla (*Figure 1*, *Figure 1—figure supplement 2*) (*Ward et al., 1975*; *Ware et al., 1975*; *Perkins et al., 1986*). The complex architecture of the endings of these two sensory neurons has not previously been characterized in detail. The dendrites of four additional sets of non-ciliated neurons, URA, URB, URX, and URY also terminate in this region but the anatomy of their endings has also not yet been extensively characterized.

## Organization of glial support cell processes within sensilla

In addition to identifying individual neuronal endings, we also identified all glial support cells present in the worm nose and reconstructed 3D models of glial processes present in the amphid, inner labial, outer labial, and cephalic sensilla (*Figure 3A,A'*, *Figure 3—figure supplement 1*). These reconstructions indicate that the endings of both neurons and glia of a given sensillum interweave extensively as they project to their target lips (*Figure 3A,A'*, *Figure 3—figure supplement 1*). As shown previously, the processes of two types of glial cells associate with neuronal cilia in each sensillum (*Ward et al., 1975*; *Ware et al., 1975*; *Perkins et al., 1986*). The socket cell process ensheaths only the most distal ciliary regions (*Figure 3B,E*, *Figure 3—figure supplement 1*), whereas the sheath cell surrounds most of the proximal cilium and distal dendritic regions (*Figure 3C,D,F*; *Figure 3—figure supplement 1*). However, some amphid organ neuron cilia are embedded in the amphid sheath cell process and do not associate with

the respective socket cell process (*Figure 3—figure supplement 1*) (*Ward et al., 1975*; *Ware et al., 1975*; *Perkins et al., 1986*). The association between neuronal and glial cell endings was consistent both between left and right sides of the same animal, as well as across animals.

The expanded socket cell processes that surround the most distal ciliary segments are connected on opposite sides of the cilium by an autocellular junction (*Figure 3B,E*). The socket process also makes CeAJ connections with the hypodermis (*Figure 3—figure supplement 2A*) and its respective glial sheath cell (*Figure 3—figure supplement 2B*); the sheath cell process also makes CeAJ connections with cilia at their base (*Figure 3—figure supplement 2C*). These junctions have previously been referred to as belt junctions (*Perkins et al., 1986*), desmosomal junctions (*Ware et al., 1975*) or tight junctions (*Ward et al., 1975*). In addition to wrapping the most distal segment of the cilium, the socket cells may also be secretory as we find that their processes appear to contain large, presumably secretory vesicles (*Figure 3B*). Sheath glial cell processes are marked by the presence of large Golgi cisternae and vesicles that contain relatively electron-dense, smooth matrix material (*Figure 3C,D,F,G*) (*Ward et al., 1975*). In some regions, the vesicles appear to fuse with the sheath cell plasma membrane bordering the cilia, suggesting that matrix material may be secreted into an extracellular sheath lumen (*Ward et al., 1975*; *Perkins et al., 1986*; *Perens and Shaham, 2005*). This matrix-filled space is especially pronounced in the amphid organs (*Figure 3C*), where the amphid sheath cell process surrounds a large lumen that tapers into a narrower channel distally that is open to the environment (*Figure 3B–D*).

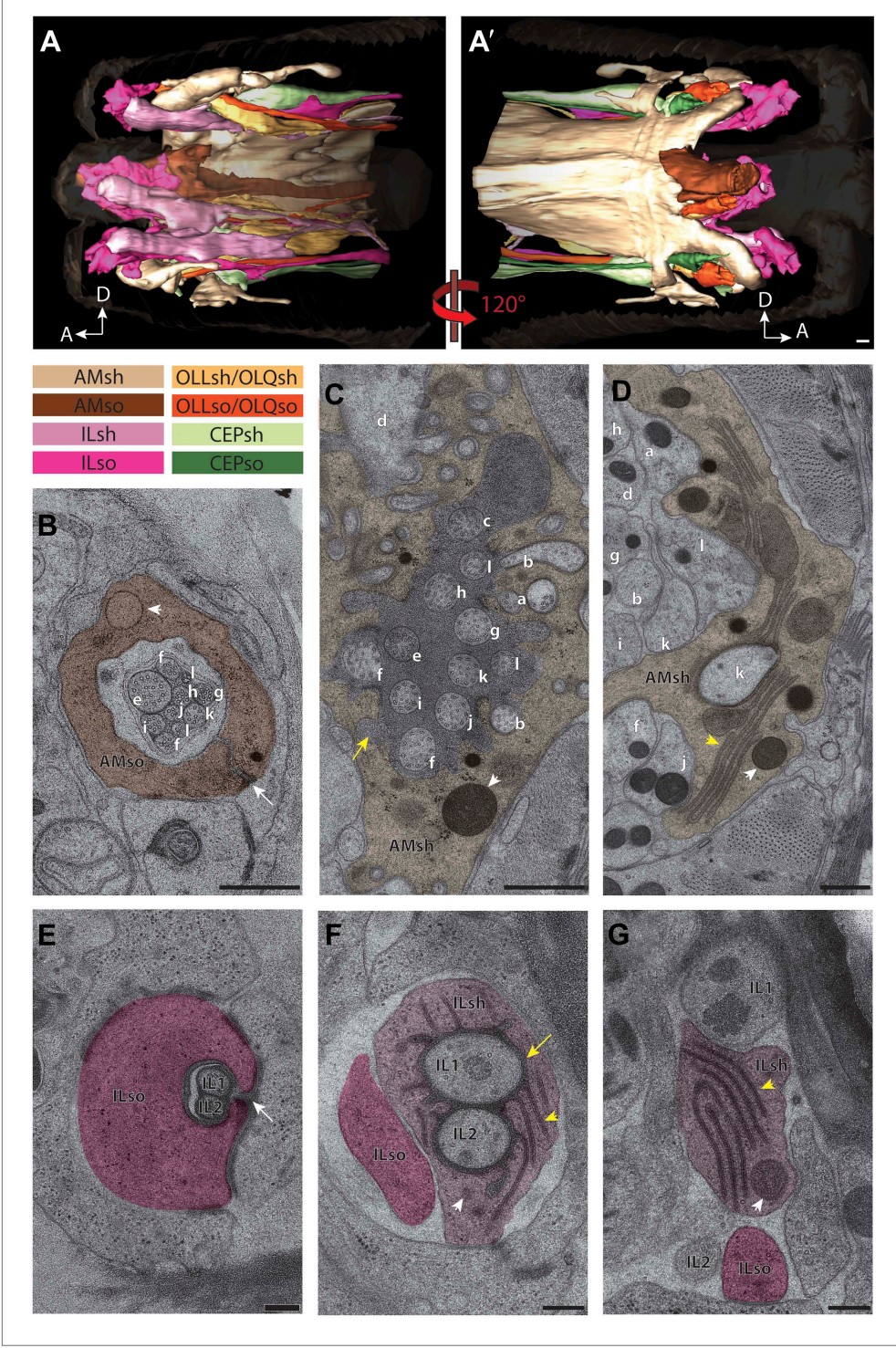

**Figure 3**. Two glial cell types are associated with each ciliated sensillum. (**A** and **A'**) 3D reconstruction models of sheath (sh) and socket (so) glia for left side of the worm for amphid (AM), inner labial (IL), outer labial (OLL/OLQ), and cephalic (CEP) glia. Glial types are color coded as indicated. See **Figure 3—figure supplement 1** for 3D models of glia associated with individual anterior sensilla. (**B**–**D**) Distal (**B**), middle (**C**) and region proximal to the amphid cilia TZ (**D**). Channel cilia of individual neurons are labeled with the last letter in their names. Arrow indicates the autocellular junction; arrowhead indicates vesicle present in the AMso process. (**C**) The AMsh glial cell process surrounds the proximal regions of amphid neuron cilia; the lumen contains electron-dense matrix (yellow arrow).
*Figure 3. Continued on next page*

*Figure 3. Continued*

White arrowhead indicates vesicle in the AMsh process. (**D**) The AMsh process contains extensive Golgi lamellae (yellow arrowhead) and secretory vesicles (white arrowhead). Cilia are labeled as above. (**E–G**) Distal (**E**), middle (**F**), and region proximal to the TZs (**G**) of the IL neurons. Autocellular junction in the ILso process is indicated by an arrow in (**E**). The lumen of the ILsh glial cell process surrounding the IL1/IL2 proximal cilia region also contains electron-dense matrix (yellow arrow in **F**), and a network of ER/Golgi (yellow arrowheads in **F** and **G**). Vesicles are indicated by white arrowheads. Scale bars: (**A–D**) 500 nm; (**E–G**) 200 nm. *Figure 3—figure supplement 2* shows additional examples of CeAJs among glial and neuronal cells in individual sensilla.

The following figure supplements are available for figure 3:

**Figure supplement 1**. 3D reconstruction models of glia associated with different sensilla.

**Figure supplement 2**. CeAJ connections between glial processes and surrounding cells.

## General ciliary ultrastructure

The prototypical cilium in adult *C. elegans* can be separated into four major segments: (a) a distal segment, (b) a middle segment that comprises the main axonemal core consisting of nine outer doublet MTs (dMTs), (c) a TZ at the base of the axoneme, and (d) a bulb-like periciliary membrane compartment (PCMC) proximal to the TZ (*Figure 4A*) (*Ward et al., 1975*; *Ware et al., 1975*; *Perkins et al., 1986*; *Scholey, 2003*; *Winkelbauer et al., 2005*; *Williams et al., 2011*; *Kaplan et al., 2012*). Subsets of specific cilia also have long or short rootlets that may be striated, and that extend proximally from the TZ often beyond the PCMC into the dendrite (*Figure 4B*) (*Ward et al., 1975*; *Perkins et al., 1986*).

In the distal segments, the axoneme typically consists of nine outer singlet MTs (sMTs), which represent extensions of the 13 pf-containing A tubules of the dMTs (*Figure 4C*) (*Ward et al., 1975*; *Ware et al., 1975*; *Perkins et al., 1986*). The dMTs of the middle segment are comprised of both the 13 pf A- and the incomplete 10 pf B-tubule that is fused to the A-tubule (*Figure 4D*) (*Linck and Stephens, 2007*; *Nicastro et al., 2011*). The transition from dMTs in the middle segments to sMTs in the distal segments manifests as the B-tubules disengage from the A-tubules, appearing as B-tubule hooks (*Figure 4—figure supplement 1A*). Although these can arise from MT instability in the middle segments (*O'Hagan et al., 2011*), our ultrastructural analyses indicate that B-tubule hooks are a common component of wild-type cilia in the middle-to-distal segment transition. In addition, both distal and middle segments can contain a variable number of inner singlet MTs (isMTs) of smaller diameter that correspond to 11-pf MTs (*Figure 4C,D*, *Supplementary file 1A*) (*Perkins et al., 1986*).

In the TZ at the base of the cilium, the nine outer dMTs are connected to the ciliary membrane by Y-shaped fibers, or Y-links (*Figure 4E*). The TZ often also contains a number of isMTs (*Figure 4E*) that sometimes appear to be attached to an inner ring-structure, known as the TZ apical ring (*Figure 4—figure supplement 1B*) (*Perkins et al., 1986*). The apical ring appears more amorphous in these HPF-FS samples than in chemically fixed sections (see *Figure 4B* in *Perkins et al., 1986*), possibly due to artifactual condensation of fibrous networks upon conventional chemical fixation (*McEwen et al., 1998*). In most organisms with cilia, the dMTs of the TZ arise from the triplet MTs of the basal body (BB), a modified centriole that gives rise to cilia (*Marshall, 2008*; *Kobayashi and Dynlacht, 2011*; *Kim and Dynlacht, 2013*). However, in *C. elegans*, BBs degenerate during the development of ciliated neurons, and thus, no obvious BB structures are observed in mature cilia (*Figure 4A*) (*Perkins et al., 1986*). Although axonemal dMTs are usually arrayed in the form of a tight cylinder in the TZ in cilia with BBs, the array of dMTs flares out at the ciliary base in *C. elegans* cilia (*Figure 4A*). We suggest that although axonemal dMTs are arranged in a tight cylindrical conformation in the axonemal shaft during ciliogenesis in *C. elegans*, this cylindrical arrangement is disrupted proximally following BB degradation. We did not observe transition fibers that usually attach to the basal body MTs (*Anderson, 1972*), at the base of any cilia in either cross- or longitudinal sections. Our ET analyses suggest that the electron-dense fiber-like structures noted previously in cross-sections (*Perkins et al., 1986*; *Williams et al., 2011*) could be dMTs that flare at the ciliary base and are viewed obliquely as a projection through a ~100-nm plastic section (*Figure 4G*, *Figure 4—figure supplement 2*). It is possible that proteins previously thought to be present at the presumptive transition fibers in *C. elegans* cilia may instead be

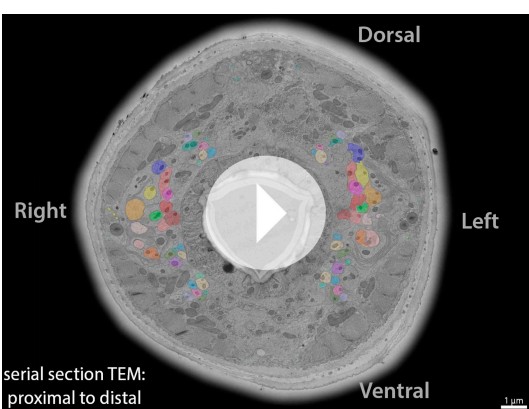

**Video 1**. ssTEM 3D reconstruction model of all ciliated and non-ciliated anterior sensory endings in the *C. elegans* adult hermaphrodite. Color codes are as indicated in *Figure 2*.

localized to the proximal flared ends of the dMTs that may be specialized, and are embedded in amorphous filamentous material (*Figure 4—figure supplement 1C*); this localization may account for their donut or V-shaped localization patterns observed previously (*Figure 4—figure supplement 1C*) (*Blacque et al., 2005*; *Williams et al., 2011*; *Wei et al., 2013*). Thus, our analyses imply that conventional transition fibers are not retained in mature *C. elegans* cilia.

Rootlets are composed of fibers comprised of the rootletin protein and are prominent electron-dense cytoskeletal structures that are sometimes striated, and that typically originate from the proximal end of a BB (*Spira and Milman, 1979*; *Wolfrum, 1992*; *Yang et al., 2002*). Rootlets are implicated in providing structural support to the cilium, maintaining ciliary integrity, stabilizing TZs and BBs, and regulating intracellular protein transport (*Yang et al., 2002, 2005*; *Bauer et al., 2011*;

*Gilliam et al., 2012*; *Mohan et al., 2013*). Only the IL1, OLQ, and BAG neurons in *C. elegans* have been reported to have striated rootlets (*Ward et al., 1975*; *Perkins et al., 1986*). ssTEM analysis of these rootlets show highly ordered striations along their length, giving them a 'muscle-like' appearance (*Figure 4B*), and a cauliflower-like cross-sectional morphology (*Figure 4F*). Consistent with degenerated BBs in the adult stage, we observed that each rootlet extends from the proximal end of the TZ and projects into the neuronal dendrite (*Figure 4B,F*). Interestingly, both IL1 and OLQ have roles in mechanosensation (*Hart et al., 1999*; *Goodman, 2006*), suggesting that the prominent striated rootlet may provide support for mechanical 'shearing stress' that the worm experiences, and/or help to transduce mechanosensory stimuli.

We observed vesicles at the base of, and/or in, several cilia, and particularly in amphid neuron cilia (*Figure 5*, *Supplementary file 1B*). As expected, many vesicles were found in the PCMC (*Figure 4—figure supplement 1C*), but we also observed vesicles between the flared dMTs at the base of several cilia (*Figure 5C*, *Supplementary file 1B*). We also detected multiple vesicles more distally at the TZ region of several amphid cilia where the dMTs form a tight cylinder (*Figure 5A,B*, *Supplementary file 1B*). These vesicles were located either within the cylinder of nine outer dMTs in the TZ (*Figure 5A*), or between the axoneme and the ciliary membrane (*Figure 5B*). With the single exception of vesicles detected in the distal segment of an AFD amphid neuron (*Figure 12B*), no vesicles were observed distal to the TZ in the ciliary shafts in our examined images. ssET analyses clearly showed the presence of a bilayered membrane in a subset of these vesicles (*Figure 5*; *Supplementary file 1B*). Although the presence of vesicles in the TZ is unexpected, it is possible that in the absence of an anchoring BB at the proximal end of *C. elegans* cilia, vesicles are able to move between the flared dMTs at the ciliary base, into the TZ region, and possibly further distally into the cilium. Vesicles and vesicle-like structures have been detected previously in the mammalian photoreceptor axoneme and in olfactory cilia (*Reese, 1965*; *Chuang et al., 2007*; *Gilliam et al., 2012*). The presence of vesicles within specialized cilia may represent an as yet understudied mechanism of ciliary protein delivery.

## Ultrastructural analyses of amphid sensory cilia

We identified each neuron within the bilateral amphid sensilla and reconstructed 3D models of their ciliated endings (*Figure 6*, *Video 2*). These neurons can be separated into three major categories based on their ciliary morphologies. Eight neurons (ASE, ADF, ASG, ASH, ASI, ASJ, ASK, ADL) contain simple rod-like 'channel' cilia that terminate within a channel formed by the amphid socket (AMso) glial cell and respond largely to aqueous compounds (*Ward et al., 1975*; *Perkins et al., 1986*; *Bargmann and Horvitz, 1991*) (www.wormatlas.org). The channel ends as a pore in the cuticle of the lateral lip, exposing the channel lumen to the environment. Three neurons (AWA, AWB, AWC) contain elaborate 'wing' cilia, which are required for responses to volatile odorants and temperature,

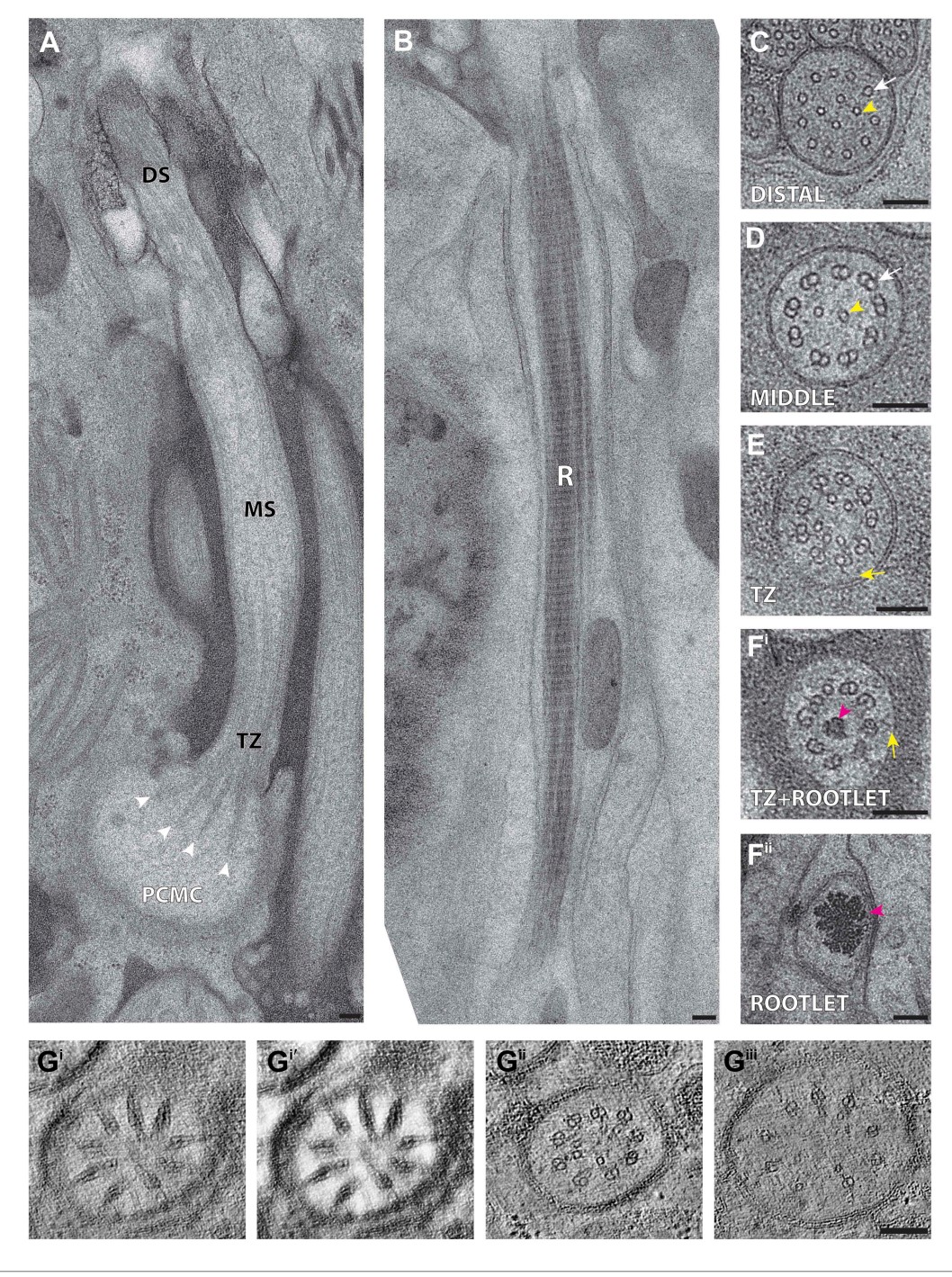

**Figure 4**. *C. elegans* ciliary ultrastructure. (**A**) Longitudinal section of an amphid channel cilium (ASE), showing distal segment (DS), middle segment (MS), TZ and PCMC. Flaring of the MTs at the ciliary base is indicated by arrowheads. (**B**) Longitudinal section of a long striated rootlet (R) within the dendrite of an IL1 neuron. Cross-section TEMs of (**C**) distal segment (ASE cilium), (**D**) middle segment (ASI cilium), and (**E**) TZ (ASI cilium). (**F**ⁱ) Cross-section TEM of the rootlet-like material within the IL1 TZ, and (**F**ⁱⁱ) the rootlet within the IL1 dendrite. White arrows indicate outer sMTs (**C**) or dMTs (**D**). Yellow arrows indicate Y-links. Yellow and pink arrowheads indicate isMTs and rootlet, respectively. (**G**ⁱ⁻ⁱⁱⁱ) ssET of flared dMTs in cross-section at the TZ of ASI cilium. (**G**ⁱ) Binned serial section tomogram (140 slices) and (**G**ⁱ') filtered image simulating a thick plastic section of the proximal ASI TZ. Serial section tomogram (10 slices binned) of the (**G**ⁱⁱ) distal and (**G**ⁱⁱⁱ) proximal part of the same TZ region as in (**G**ⁱ) shows

*Figure 4. Continued on next page*

*Figure 4. Continued*

distinct dMTs and isMTs. Scale bars: 100 nm. See *Figure 4—figure supplement 1* for additional views of MTs, apical rings and rootlet-like structures, and *Figure 4—figure supplement 2* for further ET analyses of the flared dMTs at the ciliary base.

The following figure supplements are available for figure 4:

**Figure supplement 1**. Subcellular features of amphid cilia.

**Figure supplement 2**. ET analyses of flared dMTs at the ciliary base.

and one thermo- and gas-sensing neuron (AFD) which ends in a finger-like cilium and multiple micro-villi that do not contain MTs (*Ward et al., 1975*; *Perkins et al., 1986*; *Bargmann et al., 1993*; *Mori and Ohshima, 1995*) (www.wormatlas.org). Although the distal ciliary segments of AWA, AWB, and AWC are embedded within the amphid sheath (AMsh) process and do not enter the AMso channel, their PCMCs, TZs, and proximal ciliary middle segments are exposed to the matrix-filled lumen encompassed by the AMsh cell process, which is continuous with the channel lumen, and thus in contact with the environment (*Figure 6B–D*, *Video 2*) (*Ware et al., 1975*). In contrast, the endings of the AFD neurons appear to be fully embedded within the AMsh process without contact with the matrix-filled lumen (*Figure 6B–D*; *Video 2*) (*Ware et al., 1975*). Reconstruction of the 3D models

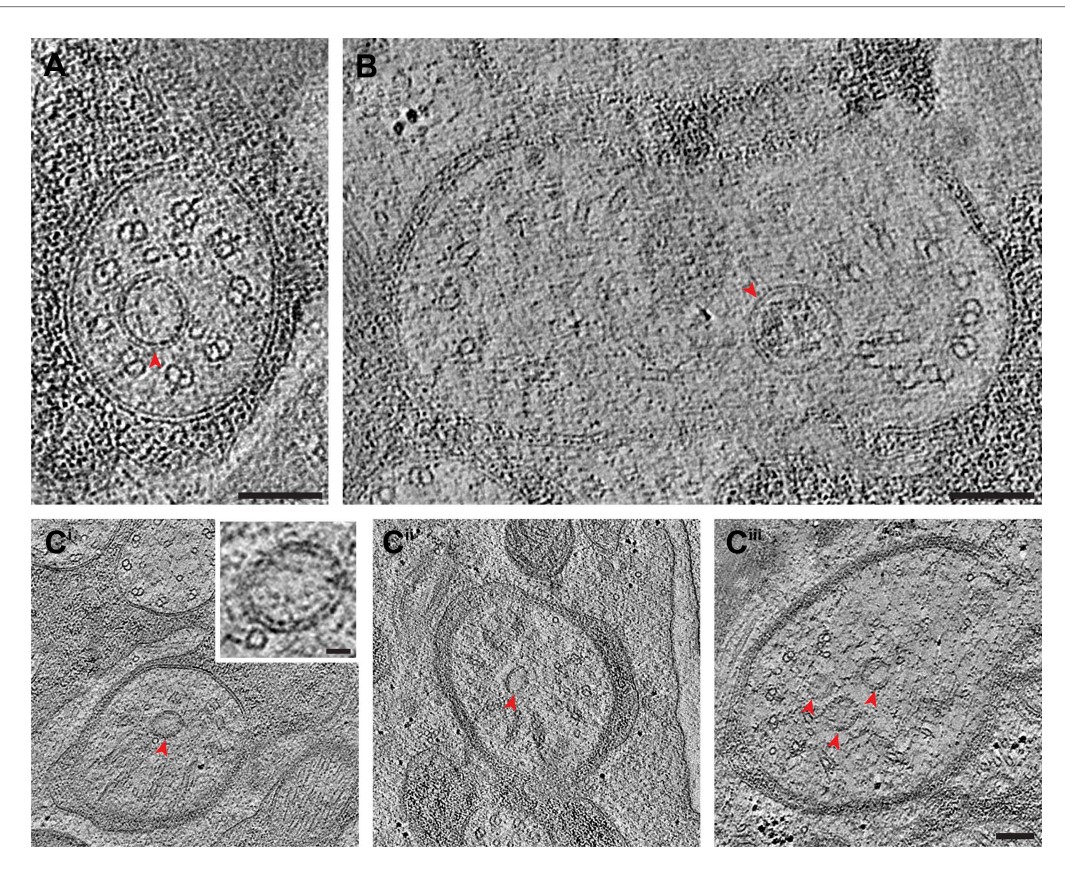

**Figure 5**. Vesicles at the ciliary base and in the TZ. Cross-section ET slices showing vesicles (red arrowheads) present (**A**) at the cylindrical region of the TZ in the ASI amphid neuron cilium, (**B**) between the axoneme and ciliary membrane of one of the two AWB amphid neuron cilia, and (**C**) between the flared dMTs at the base of ASG (**Cⁱ**), ASJ (**Cⁱⁱ**), and ASE (**Cⁱⁱⁱ**) amphid neuron cilia. The bilayer membrane of the vesicles is visible in most examples. Scale bars: (**A–C**) 100 nm (**Cⁱ** inset) 20 nm.

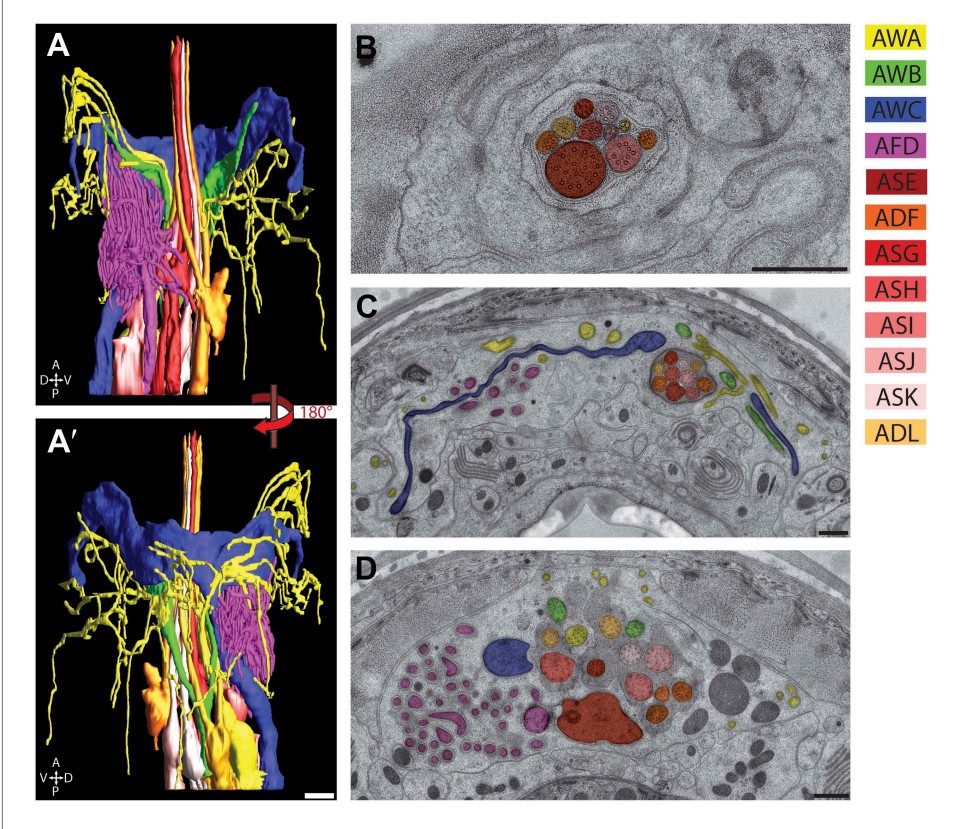

AWA
AWB
AWC
AFD
ASE
ADF
ASG
ASH
ASI
ASJ
ASK
ADL

**Figure 6**. The amphid sensillum. (**A** and **A'**) 3D graphical model of the reconstructed amphid sensilla on the right side indicating the endings of twelve sensory neurons. Scale bar: 1 µm. (**B**) Distal, (**C**) middle, and (**D**) proximal TEM cross-sections of an amphid sensillum. Endings of individual amphid neurons are color coded as indicated. Scale bars: 500 nm.

clearly confirms that amphid cilia weave around and are entwined with each other in a remarkably stereotypical manner (*Figure 6A*, *Video 2*) (*Ware et al., 1975*; *Perkins et al., 1986*). For example, the proximal dendrites of the AWB olfactory neuron type are positioned in the middle of the amphid dendrite fascicle but their cilia branch out laterally, weaving around the cilia of the AWA, AWC, and ADL amphid neurons (*Figure 6*, *Video 2*). Similarly, although the ADL and ADF dendrites are located at distinct positions in the amphid bundle, their cilia are closely appositioned in the distal amphid pore (*Figure 6*, *Figure 7*).

## Rod-like amphid channel cilia

Of the eight channel cilia, six exhibit simple single rod-shaped structures (ASE, ASG, ASH, ASI, ASJ, ASK) (*Figure 7A,A'*), whereas two contain two rods each (ADF and ADL) (*Figure 7B,B'*) (*Ward et al., 1975*; *Perkins et al., 1986*). The TZs of channel cilia are placed at different positions within the channel in a cell-type specific manner, such that the distance between the most proximal and most distal TZs of the channel cilia is 3.3 ± 0.5 µm (n = 4 sides). The TZs of the ADL neurons are positioned most proximally (12.6 ± 0.4 µm from nose tip; n = 4 sides), whereas the TZs of the ASE neurons are found most distally (9.3 ± 0.2 µm from nose tip; n = 4 sides). The diameters of the axonemes also vary among specific cilia types, with ASE cilia exhibiting the largest diameter (310 ± 20 nm; n = 4 sides) in the distal segment.

The simple rod-like morphology has been considered the prototypical ciliary type in *C. elegans* (*Figure 4*) (*Ward et al., 1975*; *Perkins et al., 1986*). In the distal and middle ciliary segments of channel cilia, as expected, we observed a cylindrical array of 9 sMTs (A-tubules) and dMTs, respectively, and 1–7 isMTs (*Figure 7C–E'*); isMT numbers are variable and inconsistent both between neurons and animals (*Supplementary file 1A*). Interestingly, ssTEM and 3D modeling show that in different

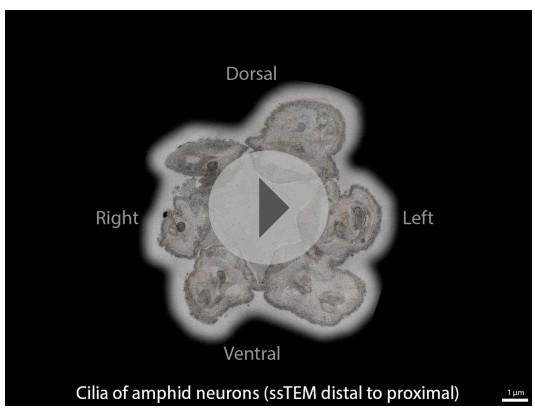

Dorsal

Right  Left

Ventral

Cilia of amphid neurons (ssTEM distal to proximal)    1 μm

**Video 2**. ssTEM 3D reconstruction model of all amphid neuron cilia and associated socket and sheath cell processes. Color codes as indicated in **Figure 2**.

cilia, the arrays of axonemal dMTs twist as a whole, with the direction of the twist reversing several times as the cilium projects distally (**Video 3**); the mechanism or consequence of this twisting is unclear but may contribute to axoneme stability. At the TZs, isMTs are often associated with a thin, fibrous apical ring just interior of the nine dMTs (**Figure 4—figure supplement 1B**) (**Perkins et al., 1986**). No striated rootlets are found in amphid neuron dendrites, although as noted previously, the ciliary base region of all channel cilia contains electron-dense rootlet-like filaments embedded in amorphous material (**Figure 4—figure supplement 1C**) (**Perkins et al., 1986**). Rootletin has recently been shown to localize to this region and regulate the integrity of amphid neuron cilia (**Mohan et al., 2013**), indicating that in *C. elegans*, rootletin is associated with cilia lacking obvious striated rootlets.

## Characterization of three types of branching at the anterior endings of sensory neurons

In contrast to the amphid single-rod channel cilia, several other amphid sensory cilia are more complex and branched. ADF and ADL contain two rod-like cilia each, whereas the AWA wing cilia exhibit highly complex higher order branches (**Ward et al., 1975**; **Perkins et al., 1986**). Overall >40% of all anterior sensory neuron endings are branched, resulting in a wide range of cilia shapes. The ultrastructure of these complex branched sensory endings, and the mechanisms that underlie this complexity, have not been studied in detail in any organism.

Based on our ultrastructural analyses, we grouped branched neuronal endings into three types characterized by the site of branching relative to the TZ of the cilium: branching at the TZ (type I), distal to the TZ (type II), and proximal to the TZ (type III) representing dendritic branching. The categorization of these branching patterns with respect to the TZ is important, because the TZ is considered to function as a diffusion barrier that regulates movement of membrane proteins into and out of the cilium (**Nachury et al., 2010**; **Czarnecki and Shah, 2012**; **Reiter et al., 2012**; **Szymanska and Johnson, 2012**). Consequently, branching distal to the TZ requires regulated transport of branch-promoting factors specific to different branching patterns across the TZ ciliary gate.

### Ciliary branching at the TZ (type I)

Ciliary branching at the TZ is observed in both double-rod channel cilia (ADF, ADL) as well as in the AWB wing cilia.

### *ADF and ADL*

We identified and modeled the ADL (**Figure 8A**; **Video 4**) and ADF (**Figure 8—figure supplement 1**) double-rod containing cilia. ADF and ADL each sends one ciliary rod to the dorsal and one to the ventral side of the channel (**Figure 7B–E'**). The lengths of the rods in each neuron type are similar (ADL—10.2 ± 0.1 μm, ADF—8.3 ± 0.1 μm; n = 4). Each ciliary rod contains the full complement of MT features found in single-rod channel cilia, including nine sMTs in their distal segments (**Figure 8B**, **Figure 8—figure supplement 1B**) and nine dMTs in their middle segments (**Figure 8C**, **Figure 8—figure supplement 1C**) along with 4–6 isMTs that originate at the TZ region (**Figure 8D,E**, **Figure 8—figure supplement 1D**, **Supplementary file 1A**) (**Ward et al., 1975**; **Perkins et al., 1986**).

The proximal origin of the nine dMTs of these cilia, and MT organization at their TZs has not been previously described. To analyze the branching pattern in detail, we reconstructed and modeled their 3D structures using both cross-sectional and longitudinal ssTEM and ssET. This analysis shows that the two ciliary rods in ADL and ADF arise from two independent TZs (**Figure 8D–F**, **Figure 8—figure supplement 1D**; **Video 4**), and are maintained as independent axonemes. The TZs of each ciliary rod are not oriented in parallel, but are instead situated at an angle to each other. The nine

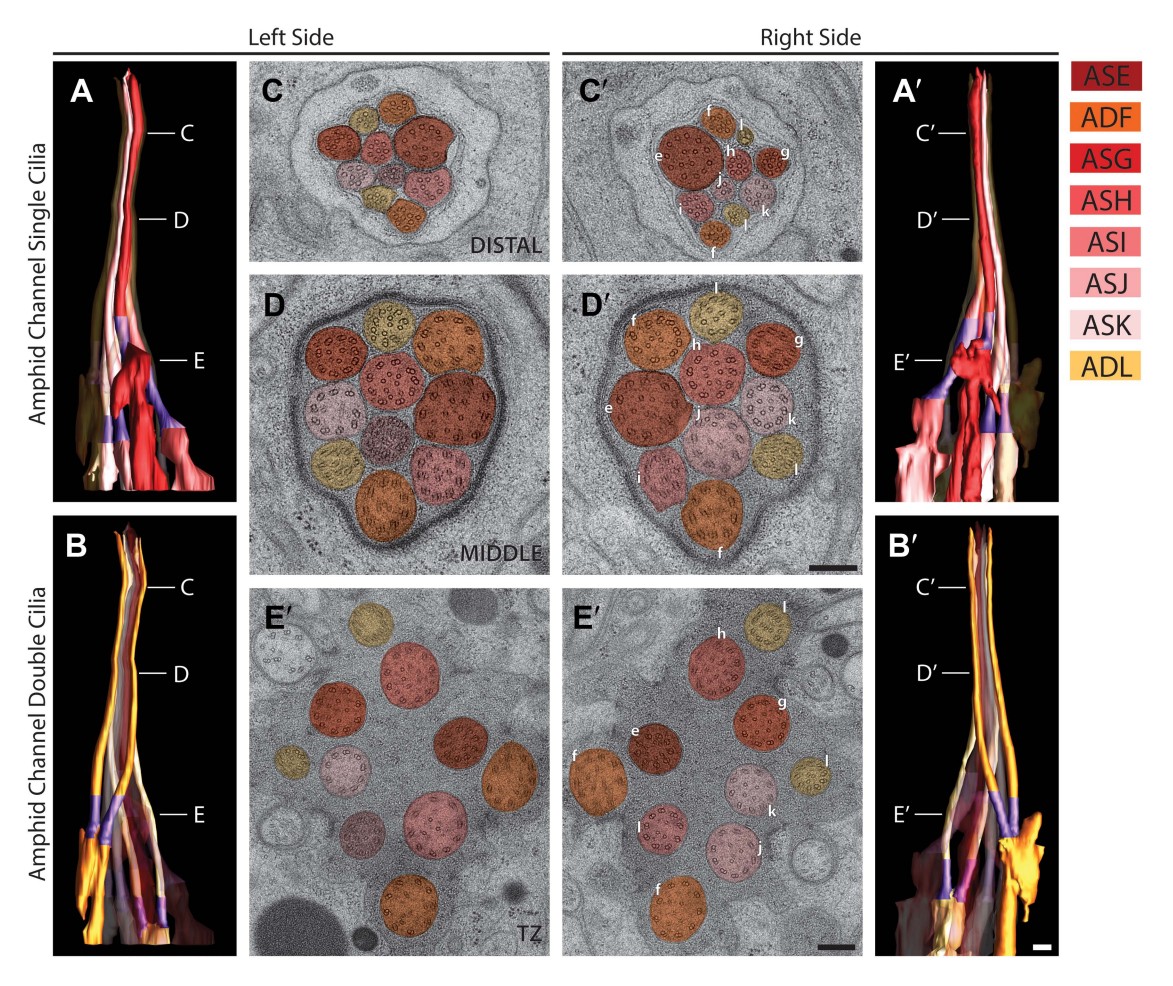

**Figure 7**. Morphology and ultrastructure of amphid channel cilia. Reconstructed 3D models of amphid channel cilia containing single (**A** and **A'**) or double (**B** and **B'**) rods on the left (**A** and **B**) and right (**A'** and **B'**) sides. Labels indicate approximate positions of cross-sections shown in **C**–**E'**. TZs (purple) are indicated. Scale bar: 500 nm. (**C**–**E'**) Cross-section TEM images of amphid channel at the level of distal segment (**C** and **C'**), middle segment (**D** and **D'**), and TZ (**E** and **E'**). Individual cilia are identified on the right only by the last letter in their names. Scale bars: 200 nm. See *Figure 3—figure supplement 2C* for additional views of the TZs and rootlet-like structures at the base of amphid neuron cilia.

Y-linked dMTs at the TZs of each ciliary rod of ADL and ADF are found within a tight ciliary shaft that flares as the ciliary base widens, so that the proximal cross-sections often show an oval array of 18 dMTs (*Figure 8D–F*, *Figure 8—figure supplement 1D*, *Video 4*).

## AWB

Each AWB neuron has two ciliary branches with variable lengths and morphologies (*Figure 9A*, *Video 2*). In particular, distal segments of these cilia exhibit wing-like membranous expansions whose extent can vary depending on environmental conditions and sensory activity (*Mukhopadhyay et al., 2008*). Although we previously reported the lack of MTs in the far distal segments using chemically fixed samples (*Mukhopadhyay et al., 2008*), we now detect sparse sMTs in the far distal portion of the AWB ciliary branches (*Figure 9B*). However, these MTs are not organized, nor are their numbers consistent from branch to branch within or between animals in either the far distal or distal segments. The MT array is more organized proximally in the middle segment, including the region where dMTs transition to sMTs (*Figure 9C,D*). However, in contrast to the 5.2 ± 0.8 µm long (averaged across all channel cilia; n = 22) middle segments of channel cilia, the middle segments of each AWB ciliary branch are short (1.7 ± 0.5 µm; n = 8). As in the case of ADF and ADL cilia, we observed that the two cilia in AWB originate from two separate TZs, each of which exhibits a well-organized, Y-linked array

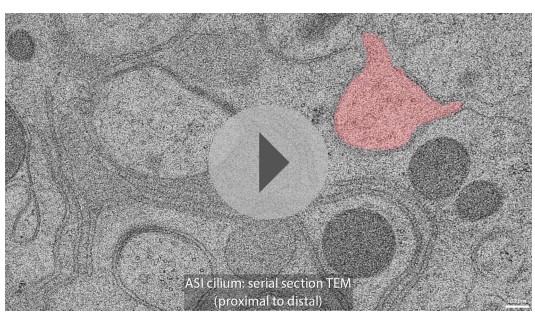

**Video 3**. 3D reconstruction model of axonemal MTs twisting as the ASI cilium projects distally. Dark purple—A-tubules; light purple—B-tubules.

of nine dMTs located at an angle to each other (*Figure 9D–F*). Thus, each ciliary branch in AWB comprises a single axoneme.

These ultrastructural analyses indicate that it is more accurate to represent ADL, ADF and AWB as containing two cilia each, as opposed to containing a doubly branched cilium. In this regard, these cilia are reminiscent of the two flagella present in the unicellular green algae *Chlamydomonas*. In both cases, the two cilia/flagella are closely juxtapositioned within a cell and originate from independent TZs. In *Chlamydomonas*, two BBs, along with two probasal bodies and an array of MTs and fibers, form a complex called the basal apparatus that apposes the bases of the two flagella (*Dutcher, 2003*; *Geimer and Melkonian, 2004*). ssTEM analyses in embryonic stages prior to BB degeneration will be necessary to determine whether a similar mechanism involving ciliary nucleation from two basal bodies, as well as their coordinated localization (*Feldman et al., 2007*), operates in *C. elegans*. However, whereas the ciliary branches in ADL and ADF are similar in length and morphology as are the two flagella in *Chlamydomonas*, the two ciliary branches in AWB can be quite heterogeneous in their ultrastructure and overall architecture. The lengths of the two flagella in *Chlamydomonas* are equalized via competition for a common pool of flagellar precursors (*Rosenbaum et al., 1969*; *Ludington et al., 2012*). To our knowledge, ciliary branches of unequal lengths have not been observed in ADL and ADF, implying that the lengths of these branches may also be regulated coordinately. In contrast, mutations in genes such as the transmembrane transporter *osta-1* gene differentially affect the lengths of the ciliary branches in AWB (*Olivier-Mason et al., 2012*), suggesting that ciliary branch lengths may be regulated independently in this neuron type.

## Ciliary branching distal to the TZ (type II)

In this type of branching, a cilium branches distal to a single TZ, that is, the neurons have a single cilium that branches subsequently from its middle and distal segments. Thus for this branching type, the molecular machinery that promotes membrane remodeling and ciliary branching must be targeted to the cilium and pass through a common TZ. We further subdivided type II branching based on whether axoneme MTs segregate into multiple ciliary branches (type IIa), or whether axoneme MTs are distributed throughout a membranous, wing-like structure (type IIb).

### AWA (type IIa)

Due to its complexity, the ultrastructure of the AWA tree-like cilium has not previously been described in detail. Through our ssTEM and 3D reconstruction methods, we identified, followed, and modeled the branches of AWA cilia (*Figure 10A*, *Video 2*). These branches spread out and project dorsally and ventrally into lips adjacent to the lateral lips that house the main ciliary projections of the amphid cilia (*Figure 10A,B*). These branches remain enclosed by processes of the AMsh cell.

Our ultrastructural analysis indicates that the branching pattern in AWA (*Figure 10C,D*, *Video 2*) is distinct from the cellular branching seen in AWB, ADF, and ADL. AWA has a single cilium that originates from a single TZ with a stereotypically ordered, Y-linked array of nine outer dMTs (*Figure 10E*) that continues into a middle segment (1.2 ± 0.1 μm; n = 4; *Video 5*). However, in the distal segment we identified higher level complex tree-like branching with as many as 80 branches (*Figure 10A*). The array of nine axonemal MTs (dMTs/sMTs) becomes disorganized and segregates haphazardly to individual branches (*Figure 10C,D*, *Video 5*). For example, the first two branches may contain four and five or other combinations of dMTs (*Video 5*). Similarly, random segregation of MTs also appears to occur in higher level branches (*Video 5*). Consequently, individual branches may contain a range of dMTs, sMTs, and/or isMT numbers; we also identified branches lacking any discernible MTs (not shown). It remains to be determined how this branching occurs, whether proteins are segregated differentially to different branches, and how protein transport occurs in branches lacking MTs.

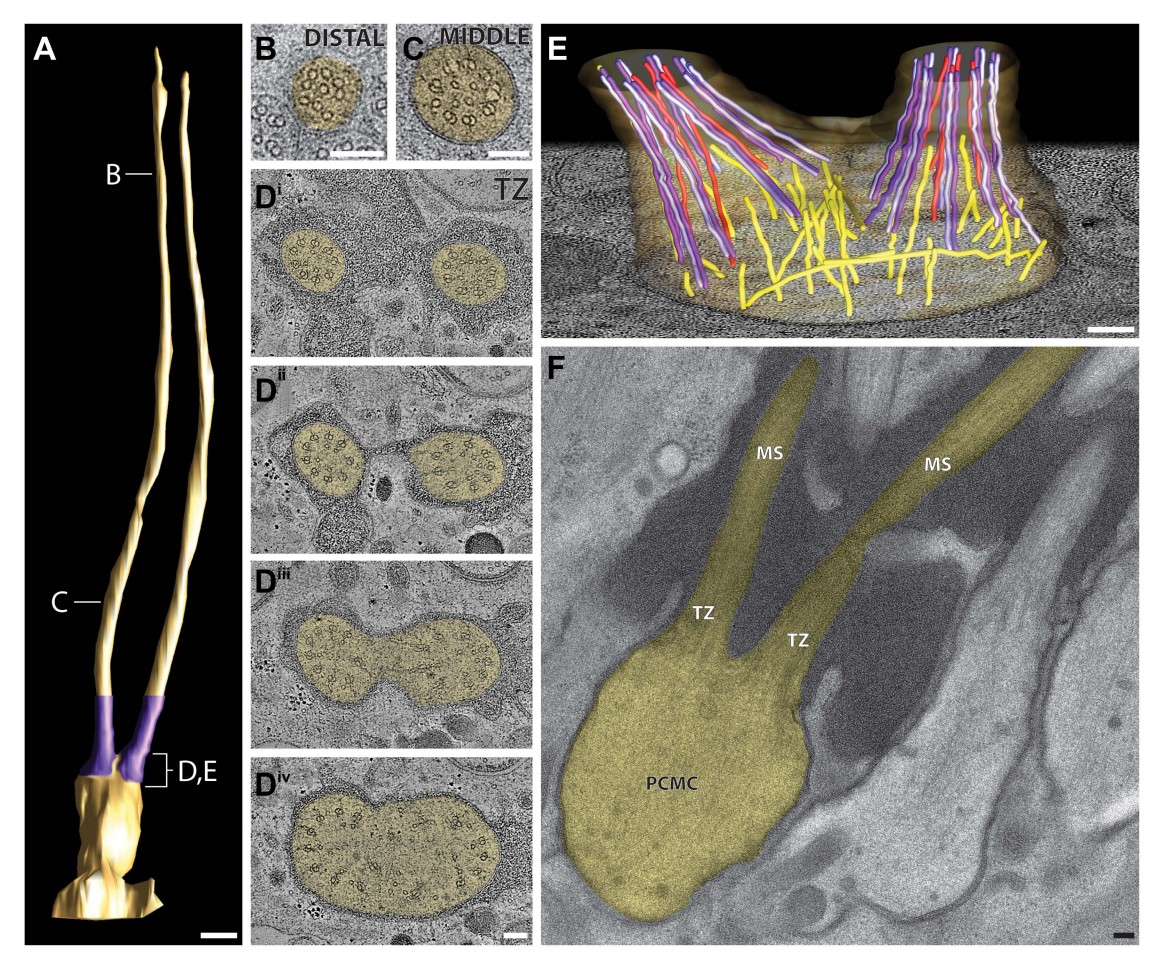

**Figure 8**. Ultrastructure of the ADL neuron cilia. (**A**) 3D reconstruction model of the two ADL channel cilia, indicating TZs (purple) at the base of each cilium. Labels indicate approximate location of sections shown in **B**–**E**. Scale bar: 500 nm. (**B** and **C**) Cross-section TEM images of distal (**B**) and middle segments (**C**) of the two ADL cilia. (**D**i–iv) ssET images of the ADL TZs (distal to proximal). (**E**) 3D reconstruction model of the ADL TZ area (indicated in **A**). Color code: dark purple—A-tubules; light purple—B-tubules; red—isMTs; yellow—other MTs in the dendrite. The ADL cell membrane is modeled in brown/tan. (**F**) Longitudinal TEM section of ADL, indicating PCMC, TZs, and middle segment (MS). Scalebars: 100 nm. See *Figure 8—figure supplement 1* for ultrastructure of ADF cilia.

The following figure supplements are available for figure 8:

**Figure supplement 1**. Ultrastructure of the ADF neuron cilia.

## AWC (type IIb)

In the AWC cilium, branching also occurs distal to the TZ but the MTs spread out into a large membranous, wing-like structure that extends into the adjacent dorsal and ventral lips and is completely embedded within the AMsh glial cell process (*Figure 6C,D*, *Figure 11A–E*, *Video 2*). Each AWC cilium has a well-ordered array of Y-link-connected nine outer dMTs and isMTs at the TZ (*Figure 11G*) (*Perkins et al., 1986*; *Evans et al., 2006*). The array of nine dMTs is retained distal to the TZ for ~1.0 μm, but the axonemal MTs are no longer organized cylindrically thereafter, and instead lie loosely side-by-side such that the dMTs spread out through the membranous elaborations (*Figure 11C–F*) (*Evans et al., 2006*). There does not appear to be a distinct middle or distal segment as observed in channel and AWB cilia. We observed sMTs and dMTs as well as isMTs in the distal and MTs in the far-distal regions of the AWC wing, consistent with previous observations (*Evans et al., 2006*) (*Figure 11A,C–E'*). The relationship between the axonemal organization and the remarkable membranous expansion is at present unclear.

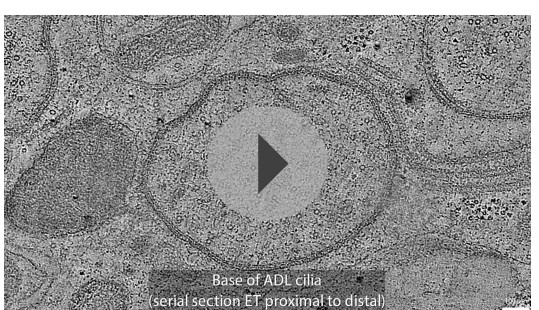

**Video 4**. 3D reconstruction model of MT distribution at the TZs of the two cilia in ADL amphid neurons. Color codes for MTs as indicated in *Figure 8* legend.

### Branching proximal to the TZ (type III)

#### AFD

The dendrite of the AFD neuron expands to form a very short unbranched cilium (1.5 ± 0.2 µm in length, 264 ± 13 nm diameter; n = 4), which extends from a large PCMC and is embedded in the center of a tuft of microvilli that emanate from the dendrite (*Figure 12A,B*, *Figure 12—figure supplement 1A*, *Video 2*). The microvilli lack MTs, are extensive, and invaginate into the AMsh cell process (*Figures 12B* and *12—figure supplement 1B*) (*Ward et al., 1975*; *Perkins et al., 1986*). We noted ribosome-like puncta in the microvilli and PCMC suggesting that these may represent sites of local translation (*Figure 12B,C*); we did not detect similar structures in other neuron sensory endings.

As in other amphid cilia, the AFD TZ forms a tight cylinder with the standard complement of nine dMTs and variable number of isMTs (*Figure 12E*). More distally, the B-tubules disengage and we observed both dMTs and sMTs in the distal regions (*Figure 12C,D*). Within the AFD ciliary shaft, we also observed amorphous electron-dense material with embedded non-striated rootlet-like structures (*Figure 12B,D*, *Figure 12—figure supplement 1B*).

In summary, amphid cilia in *C. elegans* exhibit a remarkable diversity of branching patterns. The mechanisms underlying these branching patterns are unknown, as is the functional relevance of this morphological diversity. We previously showed that neuronal activity modulates AWB ciliary morphology (*Mukhopadhyay et al., 2008*), and mutations in neuronal signaling genes lead to morphological defects in AWC and AFD sensory endings (*Roayaie et al., 1998*; *Satterlee et al., 2004*; *Mukhopadhyay et al., 2008*). Moreover, ciliary morphology is regulated via interactions with surrounding glial cells in *C. elegans* (*Bacaj et al., 2008*; *Procko et al., 2011*). These observations imply that the morphologies of sensory neuronal cilia in *C. elegans* are malleable, and can be remodeled by external stimuli and via interactions with support cells. This detailed ultrastructural description provides the foundation for further exploration of the interplay between intrinsic cell fate specification mechanisms and extrinsic signals that give rise to these complex structures.

### Amphid periciliary membrane compartments

We previously described a morphologically distinct, vesicle-containing trafficking compartment termed the periciliary membrane compartment (PCMC) at the base of *C. elegans* cilia proximal to the TZ (*Kaplan et al., 2012*). This region has previously been reported to contain multiple vesicles (*Ware et al., 1975*; *Perkins et al., 1986*). ssTEM and 3D reconstruction confirmed that the PCMC is a bulge-like region of the dendrite (*Figure 12—figure supplement 1A,B*) containing vesicles, unorganized MTs, and filamentous rootlet-like structures (*Figure 12—figure supplement 1*, *Figure 4—figure supplement 1C*). The PCMC is bound distally by the TZ and demarcated at its proximal side by electron-dense CeAJ connections between the ciliated neuron and surrounding sheath cell process.

Our 3D reconstruction revealed that the morphology of the PCMC is distinct in different neuron types (*Figure 12—figure supplement 1A,C*; data not shown). The PCMC that is positioned proximally of the two cilia in ADF is particularly bulbous (*Figure 12—figure supplement 1C*). In contrast, the PCMC of AWB is relatively small (*Figure 9F*). In AFD, the cilium and microvilli are rooted in a large PCMC, with the cilium arising distally and the microvilli branching laterally with respect to the PCMC (*Figure 12A,B*, *Figure 12—figure supplement 1A*). Interestingly, we consistently observed that an extension from the AFD PCMC protrudes into the ASE PCMC (*Figure 12—figure supplement 2*). Although the significance of this interaction is unclear, these contacts resemble the interciliary contacts observed in mammalian cells and in *Chlamydomonas*, and may represent a means of intercellular communication (*Wang and Snell, 2003*; *Ott et al., 2012*).

Since the PCMC is a site of ciliary protein trafficking (*Kaplan et al., 2012*), the distinct morphologies of PCMCs in different cilia types may correlate with different contributions and roles of

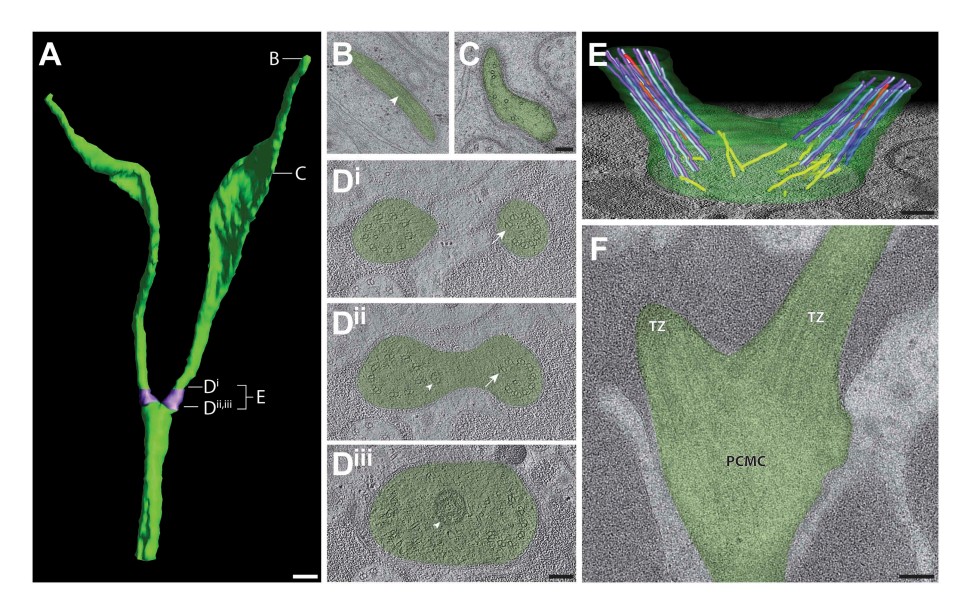

**Figure 9**. Ultrastructure of the AWB neuron cilia. (**A**) 3D reconstruction model of AWB cilia, indicating TZs (purple). Labels indicate approximate location of sections and models shown in **B**–**E**. Scale bar: 500 nm. (**B** and **C**) Cross-section TEM images of far-distal (**B**) and distal segment (**C**) of the two AWB cilia. Arrowhead indicates a MT in **B**. (**D**$^{\text{i-iii}}$) ssET images of the AWB TZs (distal to proximal). B-tubule hooks are indicated by arrows; vesicles are indicated by arrowheads. (**E**) 3D reconstruction model of the AWB TZ region. Color code: dark purple—A-tubules; light purple—B-tubules; red—isMTs; yellow—other MTs in the dendrite. The AWB cell membrane is modeled in green. (**F**) Longitudinal TEM section of AWB, indicating PCMC and TZs. Scale bars: 100 nm.

trafficking in mediating sensory neuron functions. For instance, the AFD neurons exhibit remarkable experience-dependent responses to temperature stimuli (*Kimura et al., 2004*; *Biron et al., 2006*; *Clark et al., 2006*), which could be mediated by regulating localized protein trafficking to the cilium and the elaborate microvillar region. Similarly, the ADF neurons are required for experience-dependent modulation of responses to pathogenic bacteria and familiar foods (*Zhang et al., 2005*; *Song et al., 2013*), behaviors that may also be reliant on extensive local protein turnover. It will be interesting to determine whether similar to cilia morphology in a subset of neurons, PCMC morphology and vesicle content are also dynamic and modulated by experience and context in *C. elegans*.

## Ultrastructural analyses of labial and cephalic neuron cilia

The endings of the labial and cephalic neurons extend the furthest anteriorly of all anterior sensory neurons and terminate in cuticle pores at the lip apices. These sensory neurons have been implicated largely in mechanosensation (*Kaplan and Horvitz, 1993*; *Hart et al., 1999*; *Goodman, 2006*; *Chang et al., 2011*; *Chatzigeorgiou and Schafer, 2011*; *Lee et al., 2012*).

### Cilia of the IL1 and IL2 inner labial neurons

The endings of the inner labial neurons IL1 and IL2 terminate as a juxtaposed pair at the inside of each of the six lip apices (*Figure 13A*). As reported previously (*Ward et al., 1975*; *Ware et al., 1975*; *Perkins et al., 1986*), the IL1 and IL2 cilia appear as simple rod-like cilia, however, their distal tips curve sharply while extending anteriorly such that the cilia radiate before turning back inwardly (*Figure 13A*, *Video 6*). The distal tip of IL2, whose cilium is slightly longer than that of IL1 (IL1—2.2 ± 0.2 µm; IL2—2.8 ± 0.5 µm; n = 12), zigzags again outwardly at its far distal tip (*Figure 13A*). In agreement with previous reports (*Ward et al., 1975*; *Ware et al., 1975*; *Perkins et al., 1986*), the IL1 ciliary tip is embedded and ends proximal of the IL pore within the sensillar pore cuticle, whereas the IL2 ciliary tip is exposed to the external environment through the pore that contains a cuticular plug (*Figure 13—figure supplement 1*, *Video 6*).

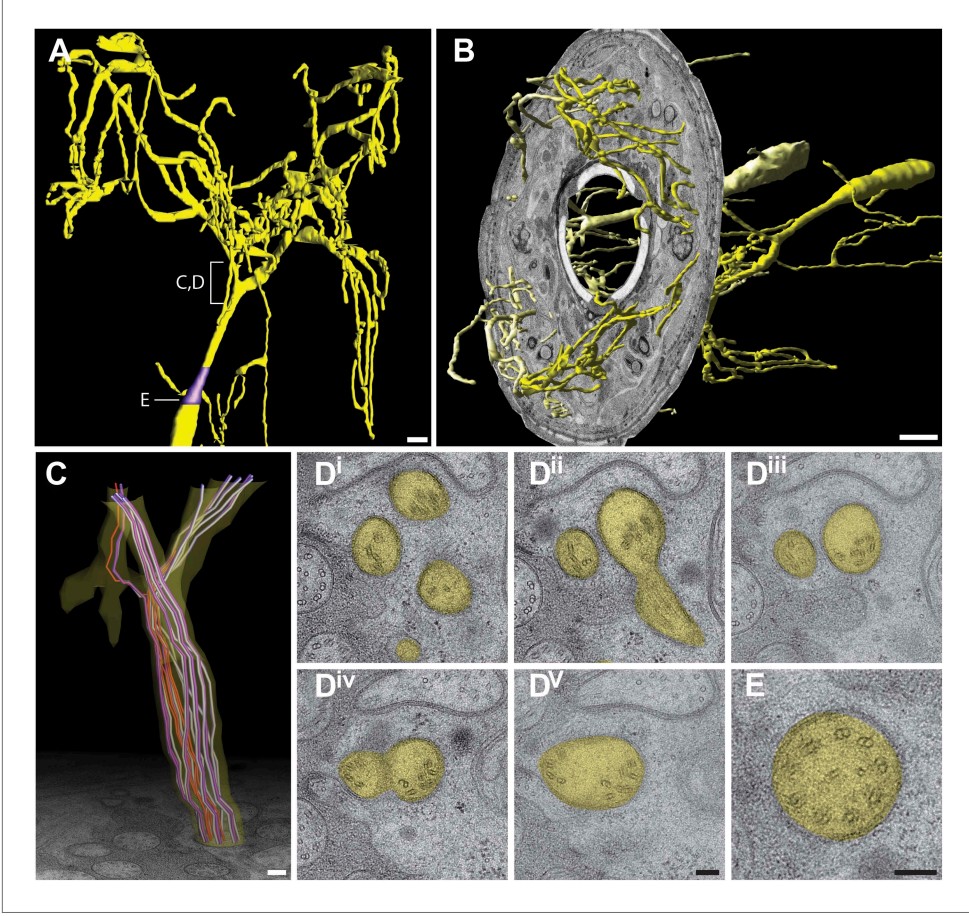

**Figure 10**. Ultrastructure of the AWA cilium. (**A**) 3D reconstruction model of an AWA cilium, indicating TZ (purple). Scale bar: 1 μm. (**B**) 3D reconstruction model of AWA cilia with a single cross-sectional view indicating the spread of AWA ciliary branches from the lateral/amphid lips to adjacent dorsal and ventral lips. Anterior at left. Scale bar: 1 μm. (**C**) Model of MTs in the region indicated in (**A**). Dark purple—A tubules; light purple—B tubules; red—isMTs. Anterior at top. (**Di–v**) Cross-section TEM series showing MT distribution upon branching from distal (**Di**) to proximal (**Dv**). (**E**) Cross-section TEM of an organized AWA TZ. Scale bar 100 nm.

Each of the IL1/IL2 neuron pairs is surrounded by a single set of sheath (ILsh) and socket (ILso) glia cells (*Figure 13—figure supplement 1*). The ILso wraps the distal segments of IL1 and IL2, and makes CeAJ connections with ILsh and the hypodermis (*Figure 13—figure supplement 1A*). The ILsh makes further CeAJ junctions with IL1 and IL2, marking the proximal boundary of the IL1 and IL2 PCMCs (*Figure 13—figure supplement 1A,B*). The lateral IL2 neurons occasionally exhibit type III dendritic branching with one to two branches emanating from their PCMC that invade the ILsh process (*Figure 13—figure supplement 1A*). We did not observe filamentous projections from the ILsh into the lumen as reported by *Ware et al. (1975)*, but observed matrix-filled invaginations into the IL2 PCMC (*Figure 13—figure supplement 1B*).

The distal segments of IL1 and IL2 each contain only a few (4–6) disorganized sMTs; we did not observe isMTs (*Figure 13B*). Electron-dense material is present in the shape of a 'disk' at the tips of IL1 but not IL2 cilia (*Figure 13—figure supplement 1B*) (*Perkins et al., 1986*); however, in contrast to previous reports (*Perkins et al., 1986*), we did not observe linkers between the disks and the ciliary membrane. Although *Ward et al. (1975)* reported seven dMTs in the TZs of IL1, we find that IL1 TZs generally contain a full complement of nine Y-linked organized dMTs (*Figure 13C*) consistent with observations by *Perkins et al. (1986)*. Unusual among all examined cilia in this study, IL2 TZs contain fewer (5–7) Y-linked dMTs (*Figure 13C*) (*Ward et al., 1975*; *Perkins et al., 1986*). We observed electron-dense material that appears to be continuous with the long

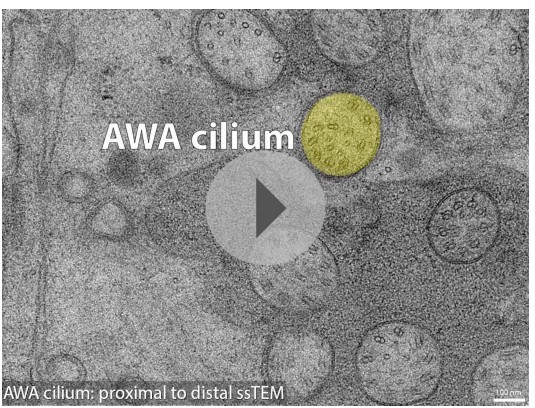

**Video 5**. 3D reconstruction model of MT distribution in the branches of the AWA cilium. Color codes for MTs as indicated in **Figure 10** legend.

(11.4 ± 0.5 µm) striated rootlets of IL1 that extend proximally from the TZs (**Figure 4B,F**, **Figure 13C,D**, **Figure 13—figure supplement 1B**) (**Ward et al., 1975**; **Ware et al., 1975**; **Perkins et al., 1986**). Interestingly, we also detected electron-dense punctae in cross-section at the TZs of IL2 that may correspond to rootlet-like structures (**Figure 13C,D**, **Figure 13—figure supplement 1B**). These punctae are particularly large in the TZs of lateral IL neurons in which long (5 µm) filamentous and non-striated structures can be observed extending from the TZ into the axoneme in longitudinal sections. The presence of these rootlets and rootlet-like structures is consistent with the demonstrated roles of both IL1 and IL2 neurons in transducing mechanosensory stimuli (**Hart et al., 1999**; **Goodman, 2006**; **Lee et al., 2012**).

## Cilia of OLL, OLQ outer labial and CEP cephalic neurons

### OLL

The two lateral outer labial OLL putative mechanosensory neurons are found bilaterally and terminate ventrally of the amphid pore (**Figure 14A–E**, **Video 6**) (**Ward et al., 1975**; **Ware et al., 1975**; **Perkins et al., 1986**; **Chang et al., 2011**). The OLL endings comprise a cilium with a TZ, but also include a type III branch from the PCMC that extends distally in parallel with the OLL cilium as an extension of the OLL dendrite (**Figure 14A**, **Figure 14—figure supplement 1A**). The OLL cilia are wrapped by the OLL sheath (OLLsh) and OLL socket (OLLso) glial cell processes (**Figure 14—figure supplement 1A**). Distal to the socket cell process, we observed an electron-dense structure we refer to as a 'spur' that may represent matrix secreted by a support cell (**Figure 14—figure supplement 1A**). In addition, an electron-dense cuticular 'string' previously referred to as an 'electron-dense branch' or a 'small dark nubbin' extends from the OLL distal tip through the cuticle (**Figure 14—figure supplement 1A**, **Video 6**) (**Ward et al., 1975**; **Ware et al., 1975**; **Perkins et al., 1986**).

As reported previously (**Ware et al., 1975**; **Perkins et al., 1986**), the distal tip of the OLL cilium is filled with electron-dense material, with few, if any, discernible MTs pushed to the cell periphery (**Figure 14B**). The distal regions also contain the electron-dense material that is more compact and disorganized with a few (3–6) MTs visible around the periphery (**Figure 14C**). The middle segment also contains few dMTs/sMTs (**Figure 14D**), whereas the TZ has a full complement of nine dMTs with 1–3 isMTs (**Figure 14E**). We did not observe a striated rootlet.

### OLQ

Each of the four quadrant lips contains closely associated OLQ and CEP sensilla (**Figure 14F**, **Video 6**). Both OLQ and CEP neurons have been implicated in mechanosensation (**Kaplan and Horvitz, 1993**; **Hart et al., 1999**; **Sawin et al., 2000**; **Kindt et al., 2007a**, **2007b**; **Kang et al., 2010**; **Chatzigeorgiou and Schafer, 2011**). Like the glia in other sensilla in the worm head, the process of each OLQ socket (OLQso) glia wraps around the distal segments of OLQ cilia, however, the far distal segment of OLQ cilia extends further and is embedded in the cuticle (**Figure 14—figure supplement 1B**) (**Ware et al., 1975**). The OLQ sheath glia (OLQsh) process wraps around the proximal cilium, PCMC and TZ of OLQ. Similar to OLL, distal to the socket cell process, there is a spur (**Figure 14—figure supplement 1B,D**) and a string that projects into the cuticle (**Figure 14—figure supplement 1B,D**); however, compared to OLL where the string is located at the very distal tip of the dendrite, the OLQ string originates more proximally from the distal segment (compare **Figure 14—figure supplement 1A,B**, **Video 6**) (**Ward et al., 1975**; **Ware et al., 1975**; **Perkins et al., 1986**).

Morphologically, the OLQ cilia have a simple rod-like shape and contain electron-dense material in their far-distal segments similar to the OLL cilia (**Figure 14G**), but also exhibit unique ultrastructural

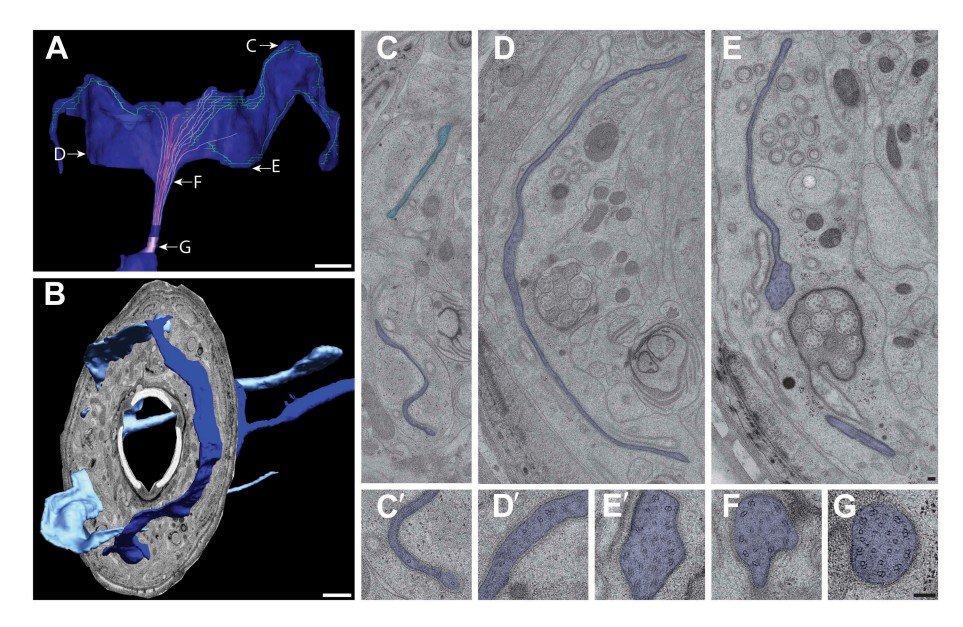

**Figure 11**. Ultrastructure of the AWC cilium. (**A**) 3D reconstruction model of an AWC winged cilium, indicating TZ (purple) and MTs. Light purple—dMTs; red—isMTs; green—MTs (undetermined). Labels indicate approximate location of sections shown in **C–G**. Scale bar: 1 µm. (**B**) 3D reconstruction model of left and right AWC cilia with a single cross-sectional view indicating the extension of the AWC wings from their lateral/amphid lip to adjacent dorsal and ventral lips. Anterior is at left. Scale bar: 1 µm. (**C** and **C′**) Distal segment of AWC indicating few sMTs. In lighter blue is the AWC cilium from the opposite side. (**D** and **D′**) Middle segment of AWC indicating few sMTs and dMTs. (**E** and **E′**) Proximal middle segment of AWC indicating spreading of the AWC wing and disorganized isMTs and dMTs. (**F**) dMTs and isMTs become disorganized in the middle segment of AWC. (**G**) The AWC TZ with nine dMTs and isMTs. Scale bar: 100 nm.

features (*Figure 14G–K*). In earlier studies, the OLQ cilia MTs were described as unusually dark and potentially filled with electron-dense material (*Ward et al., 1975*; *Perkins et al., 1986*). In this study, we clearly observed that both the A- and B-tubules of all OLQ cilia dMTs contain electron-dense material inside the MT lumen throughout the entire cilium length (*Figure 14H–J*). ssET revealed that these electron-dense structures are filamentous (*Figure 14H–J*), rather than globular and periodic as described for most MT inner proteins (MIPs) or luminal particles in MTs (*Garvalov et al., 2006*; *Sui and Downing, 2006*; *Nicastro et al., 2011*; *Schwartz et al., 2012*; *Topalidou et al., 2012*). The OLQ MIPs instead appear to resemble the filamentous 'beak'-MIPs of unknown function previously described in the B-tubules of selected dMTs in *Chlamydomonas* flagella (*Hoops and Witman, 1983*; *Bui et al., 2009*; *Nicastro et al., 2011*).

In the distal segments of OLQ cilia, four MIP-containing dMTs form a stereotypical and unique square pattern around a central filament when viewed in cross-section (*Figure 14H*). The dMTs appear to be connected to each other via thin fibers both between neighboring dMTs and through the central density to dMTs on the opposite side (*Figure 14H*) (*Ward et al., 1975*; *Ware et al., 1975*; *Perkins et al., 1986*). This is reminiscent of the MT cross-linkers observed in insect campaniform mechanoreceptors (*Liang et al., 2013*). In the middle segment, the nine MIP-containing dMTs are somewhat disorganized in the OLQ (*Figure 14I*), but form a well-organized cylinder of nine Y-linked dMTs in the TZ (*Figure 14J*). OLQ also has a short striated rootlet (2.5 ± 1.2 µm; n = 7) that extends from the TZ proximally (*Figure 14K*), as described previously (*Ward et al., 1975*; *Ware et al., 1975*; *Perkins et al., 1986*).

## CEP

Each mechanosensory CEP neuron is closely associated with an OLQ neuron throughout its dendritic endings (*Figure 14F*, *Video 6*) (*Ward et al., 1975*; *Ware et al., 1975*; *Perkins et al., 1986*). Most of

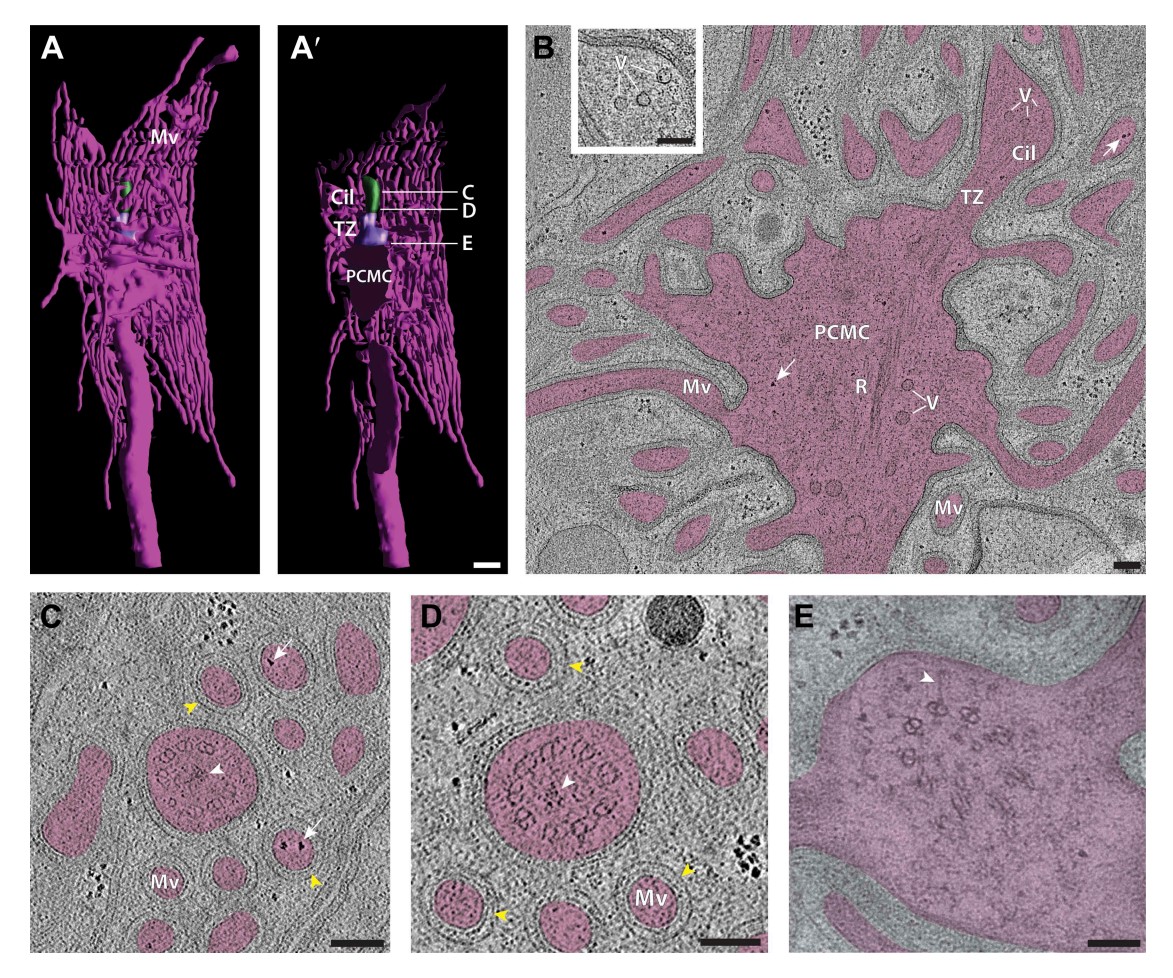

**Figure 12**. Ultrastructure of the sensory endings of AFD. (**A** and **A'**) 3D reconstruction model of an AFD finger cell sensory ending with microvilli branches (magenta) and cilium (green). (**A'**) Clipped view allowing visualization of the TZ (purple) and PCMC. Scale bar: 500 nm. (**B**) Longitudinal ET slice of the AFD sensory ending indicating cilium (Cil), TZ, PCMC, and microvilli (Mv). Vesicles (V) and rootlet-like structures (R) are also marked. Ribosome-like structures are indicated by arrows. (**C** and **D**) Cross-sectional ET slice of AFD cilium distal-middle segment (**C**) and proximal-middle segment (**D**). Ribosome-like structures are indicated by white arrows; electron-dense rootlet-like material is indicated by white arrowheads; rings of regularly arranged particles of unknown composition between the AMsh and AFD cell membranes are visible (yellow arrowheads). (**E**) Cross-section TEM of the AFD TZ indicating long Y-links (white arrowhead). Scale bars: 100 nm. *Figure 12—figure supplement 1* shows additional views of AFD morphology and ultrastructure. *Figure12—figure supplement 2* shows interaction between the PCMCs of AFD and ASE.

The following figure supplements are available for figure 12:

**Figure supplement 1**. Periciliary membrane compartments.

**Figure supplement 2**. Interactions between the PCMCs of AFD and ASE.

the CEP distal ciliary ending is wrapped by the CEP socket (CEPso) glial cell process, and similar to the OLQ distal ending, the distal tip of the CEP cilium protrudes further anteriorly and is embedded in the cuticle (*Figure 14—figure supplement 1C*). The CEP sheath (CEPsh) glial cell process wraps the CEP TZ and PCMC but does not wrap around the dendrite proximal to the PCMC (*Figure 14—figure supplement 1C*). Unlike other glial cells in the head, the processes of the CEP glia also envelop the nerve ring, the major neuropil in the head of *C. elegans* (*Ward et al., 1975*; *White et al., 1986*). Similar to OLQ, the distal segments of CEP are associated with a spur and an electron-dense string that projects into the cuticle (*Figure 14—figure supplement 1C,D*, *Video 6*) (*Ward et al., 1975*; *Ware et al., 1975*; *Perkins et al., 1986*).

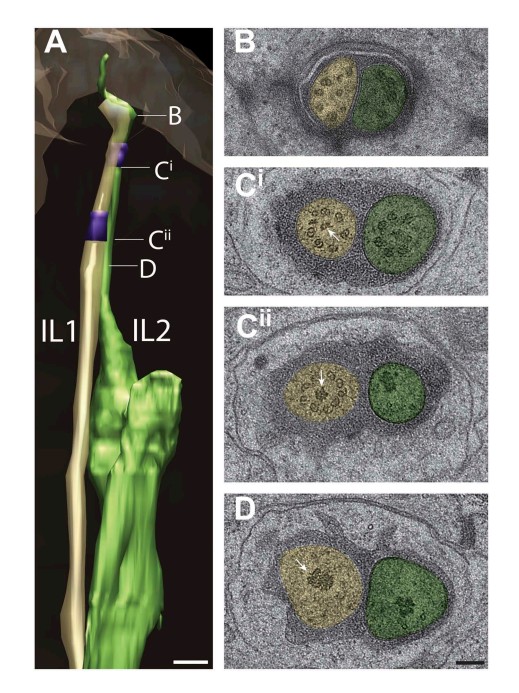

**Figure 13**. Ultrastructure of IL1 and IL2 neurons. (**A**) 3D reconstruction model of lateral IL1 (yellow-tan) and IL2 (green) cilia indicating TZs (purple) and the long striated IL1 rootlet (white). Labels indicate approximate positions of cross-sections shown in **B**–**D**. (**B**–**D**) Cross-section TEM images of IL1 and IL2 at the distal segment which has few, disorganized sMTs (**B**), IL1 (**C**ii) and IL2 (**C**i) TZs which include portions of the IL1 rootlet-like material (arrows). Scalebar: 100 nm. See **Figure 13—figure supplement 1** for additional ultrastructural features present in IL1 and IL2 cilia.

The following figure supplements are available for figure 13:

**Figure supplement 1**. Structural features associated with lateral IL1 and IL2 cilia.

Morphologically, the CEP cilia have a simple rod-like shape (**Figure 14F**) similar to OLQ (**Figure 14F**). The far distal end of CEP widens and contains a tubular body that is a cluster of numerous tubular body singlet MTs (tbsMTs) that are surrounded or connected by an amorphous, electron-dense, tubule-associated material (**Ward et al., 1975**; **Ware et al., 1975**; **Perkins et al., 1986**) (**Figure 14L**). We find that the >30 tbsMTs have a larger diameter than the 13-pf A-tubules of the dMTs, suggesting that they consist of 14–15 pfs (**Figure 14L**). Similar tubular bodies have been observed in mechanosensory sensilla, for example, the tubular bodies of insects (**Keil, 1997**). Proximal to the ciliary tip, the tbsMTs and tubule-associated material are absent, and the tbsMTs do not appear to be connected to the axonemal dMTs or sMTs (**Figure 14M,N**). The middle segment of CEP contains approximately five dMTs and few isMTs (**Figure 14M**), whereas the CEP TZ contains the typical array of nine Y-linked dMTs and several isMTs (**Figure 14N**). CEP cilia do not appear to contain a striated rootlet (**Ward et al., 1975**; **Ware et al., 1975**).

## Ultrastructural analyses of FLP and BAG cilia

The sensory endings of BAG and FLP neurons exhibit a highly complex 3D architecture, which has not previously been characterized in detail. As reported previously (**Ward et al., 1975**; **Perkins et al., 1986**), the sensory endings of BAG and FLP are not wrapped by their own set of support glia cells. Instead, there appears to be a role reversal such that the neuronal dendrites wrap or engulf part of the process of the ILSo cell associated with the lateral IL sensilla (**Figure 15**, **Video 7**) (**Ward et al., 1975**; **Ware et al., 1975**).

### BAG

The gas-sensing BAG neuron (**Zimmer et al., 2009**; **Bretscher et al., 2011**; **Hallem et al., 2011**) is named after its unique bag-like ciliary morphology (**Ward et al., 1975**; **Perkins et al., 1986**). The BAG neuron projects within the lateral fascicle together with the lateral IL and OLL sensilla (**Figure 1**, **Figure 1—figure supplement 2**, **Video 7**). Proximally, the BAG cilium contains a TZ with a typical cylindrical array of nine Y-linked outer dMTs (**Figure 15E**) and a short previously described striated rootlet (3.6 ± 0.4 μm; n = 4) (**Figure 15F**) (**Ward et al., 1975**; **Ware et al., 1975**; **Perkins et al., 1986**). BAG appears to lack a clear PCMC. The BAG dendrite projects distally, where its ciliary compartment expands and forms two to three type II ciliary branches that wrap seamlessly (i.e., continuous membrane without CeAJs or autocellular junctions) around finger-like ILso projections (**Figure 15B**). The dMTs are distributed haphazardly to the BAG ciliary branches (**Figure 15C,D**), reminiscent of the MT segregation pattern in the AWA ciliary branches. One ciliary branch further expands to 'cuff' the ILso process resembling a horseshoe-shaped structure in cross section (**Figure 15C**). This expansion is accompanied by splitting of MTs into two clusters that segregate to opposite ends of the horseshoe (**Figure 15C**). The dMTs within the BAG branches resolve to sMTs distally (**Figure 15C**).

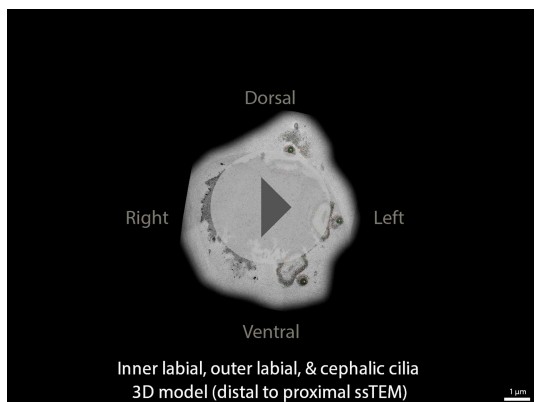

**Video 6**. 3D reconstruction model of the sensory endings and associated glial cells processes of the IL, OLL/OLQ and CEP neurons.

### FLP

The dendritic endings of the mechano- and thermosensory FLP neurons (*Kaplan and Horvitz, 1993*; *Chatzigeorgiou et al., 2010*; *Albeg et al., 2011*; *Chatzigeorgiou and Schafer, 2011*; *Liu et al., 2012*) originate from the dorsal fascicle that includes the dorsal IL, OL, and CEP sensilla. The distal dendrite of FLP then detours ventrally towards the lateral IL sensillum (*Figure 1*, *Figure 1—figure supplement 2*, *Video 7*), where 'flap'-like ciliary elaborations of FLP form a close neuron-glial relationship with ILso glial cell processes (*Figure 15B,G*, *Video 7*) (*Ward et al., 1975*; *Perkins et al., 1986*). The distal dendrite of FLP expands to a broad PCMC, which constricts to a TZ containing nine Y-linked outer dMTs and rootlet-like structures (*Figure 15I–J*). Immediately distal to the TZ, the FLP ciliary membrane expands with the axonemal dMTs segregating randomly to both sides (*Figure 15G–H*). At the distal ends, the FLP ciliary flaps surround an ILso glial cell elaboration with the MTs randomly segregating to different sides of the flaps (*Figure 15G,H*).

Proximal to the cilium of the primary FLP dendritic branch, the FLP dendrite branches to form an intricate and complex network of multi-order branches that arise during late larval development (*Ward et al., 1975*; *Albeg et al., 2011*), but that have not been characterized previously by ssTEM, particularly in the nose. The higher order branches also project distally and innervate each of the six labial lips (*Figure 15A*) but lack axonemal MTs. Furthermore, at the very distal endings we identified quaternary branches that often project perpendicular to the anterior–posterior axis and run along the cuticle or muscle cells (*Figure 15A*). Our ssTEM and electron tomograms show that the FLP dendritic branches contain unique iterative bulb-like structures (length—1.1 ± 0.1 μm; n = 5), where each bulb is separated by a constriction that can be as narrow as ~60 nm, tightly constraining the 2–3 sMTs contained therein (*Figure 15B,K*).

Among anterior sensory neurons, BAG and FLP are unusual in that they are not present in a sensillum together with their own set of glial cells, but instead associate with only the socket cells of the lateral IL sensilla. It is interesting to note that upon ablation of their cognate socket cell precursors in embryonic stages, both CEP and OLL neurons were shown to become associated with ILso glia (*Sulston et al., 1983*). These observations indicate that neurons and glia of different sensilla can interact and that moreover, socket cells may actively attract neurons of the cognate or neighboring sensilla. These results suggest that association of BAG and FLP sensory endings with ILso processes may simply provide structural stability, although we cannot rule out the possibility that BAG and FLP also communicate with the lateral IL neurons via ILso processes.

## Ultrastructural analyses of the sensory endings of non-ciliated URX and URY neurons

In addition to the ciliated anterior sensory neurons described above, the dendritic endings of four non-ciliated neuron types have been described to project with the anterior sensory fascicle bundles (*Figure 16A*). These include projections from four URA and two URB neurons of unknown function found dorsally and ventrally, two dorsal URX gas-sensing neurons (*Zimmer et al., 2009*; *Bretscher et al., 2011*; *Busch et al., 2012*) and four (two dorsal and two ventral) URY neurons that are involved in mate-searching behavior in males and may also be implicated in pathogen sensing (*Pradel et al., 2007*; *Barrios et al., 2012*) (*Figure 16A*). Sensory endings of the URA and URB neurons were only present in a few sections in the region examined here and were not modeled further.

### URX

Processes from the two URX neurons project anteriorly into the subdorsal labial process bundles. The dendritic endings are not ciliated and terminate in flat, membranous and branched elaborations

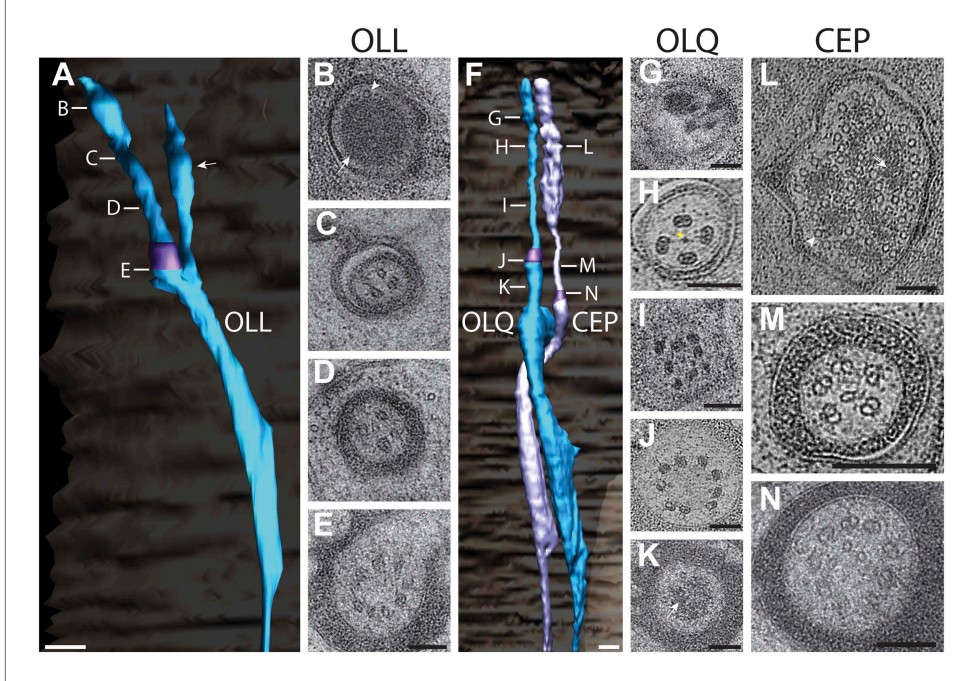

**Figure 14**. Ultrastructure of the OLL/OLQ and CEP neuron cilia. (**A**) 3D reconstruction model of OLL cilium indicating TZ (purple) and a non-ciliary type III dendritic branch (arrow). Approximate locations of cross sections in **B**–**E** are shown. Scale bar: 500 nm. (**B**) Far distal segment of OLL containing electron-dense material (arrow) and MTs (arrowhead). (**C**–**E**) Distal (**C**) and middle segments (**D**) of OLL have few disorganized MTs whereas the OLL TZ (**E**) has nine dMTs. Scale bars: 100 nm. (**F**) 3D recconstruction model of CEP (lavender) and OLQ (blue) cilia indicating TZs (purple). Approximate locations of cross sections in **G**–**N** are shown. Scale bar: 500 nm. (**G**) Cross section of OLQ far distal segment containing electron-dense material. (**H**) Cross-sectional ET slice shows distal segment of OLQ containing four dMTs linked by thin fibers to each other and to a central filament (arrowhead); note the electron-dense MIPs in both the A- and B-tubules of all dMTs (seen as black dots in the tubule-lumen). (**I**–**J**) Cross-sectional TEM (**I**) and ET (**J**) images of the disorganized middle segment (**I**) and at the OLQ TZ (**J**). (**K**) OLQ has electron-dense material (arrow) corresponding to a striated rootlet that is directed from the OLQ TZ into the dendrite. (**L**) Cross-sectional ET slices at the distal end of the CEP sensory ending shows a large tubular body containing many MTs (arrowhead) together with amorphous, electron-dense material (arrow). (**M**) dMTs and sMTs present within the middle segment of a CEP cilium. (**N**) Cross-section TEM image shows nine dMTs with isMTs in the CEP cilia TZ. Scale bar: 200 nm. See *Figure 14—figure supplement 1* for views of the spurs and cuticular strings associated with OLL/OLQ and CEP cilia.

The following figure supplements are available for figure 14:

**Figure supplement 1**. Structural features associated with OLQ and CEP cilia.

around the processes of the dorsal ILso cells (*Figure 16A,B*). We observed 2–5 single MTs in both distal and proximal cross-sections (*Figure 16B*).

## URY

URY dendrites project into the two subdorsal and two subventral labial process bundles. URY dendrites also terminate in membranous non-ciliary elaborations of a larger area than those present at URX endings. As in URX, these endings are also branched; the exact morphologies of the endings of the four URY neurons are variable but are apposed to the processes of the ILshD and OLQshD cells. We observed 4–6 single MTs in the main branches, which segregate into the distal branches, with individual branches containing 1–2 MTs. In 2 of 5 examined animals, we also observed a URY dendritic branch extending across the midline boundary into the adjacent quadrant, possibly allowing interactions with other sensilla (not shown).

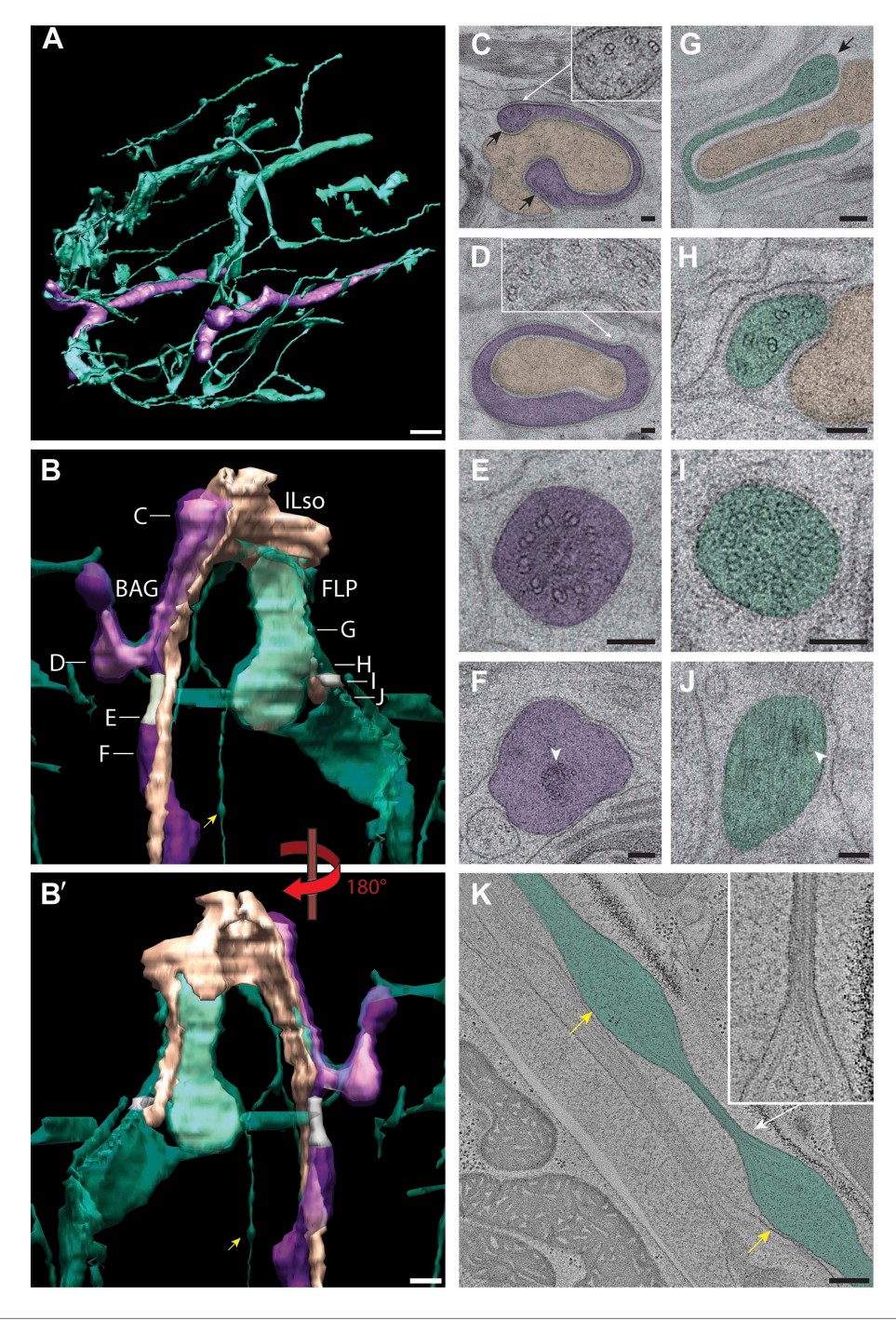

**Figure 15**. Ultrastructure of BAG and FLP cilia. (**A**) 3D reconstruction model of BAG (purple) and FLP (green) sensory endings. BAG and FLP sensory endings are positioned laterally. FLP has an extensive dendritic branching network. Scale bar: 1 μm. (**B** and **B'**) 3D reconstruction models of BAG and FLP endings indicating TZs (gray). BAG and FLP (transparent) closely ensheath the ending of the ILso glial cell (yellow-tan). Approximate locations of sections shown in **C**–**J** are indicated. Arrowheads indicate bulbous structures in FLP dendritic branch. (**C**) Cross-section TEM image showing the far distal end of BAG cilium with several dMTs (inset) at the edges of the BAG ciliary branches (black arrows) and associating with the ILso process. (**D**) The distal segment of the BAG cilium wraps around a projection of the ILso and contains dMTs (inset) segregating preferentially to one side. (**E** and **F**) Ultrastructures of the BAG TZ (**E**) and rootlet (arrowhead in **F**). (**G**) The distal segment of FLP

*Figure 15. Continued on next page*

*Figure 15. Continued*

cilia has flap-like projections with several dMTs (arrow). (**H**) The middle segment of FLP cilia contains several disorganized dMTs. (**I**) The FLP TZ contains nine dMTs. (**J**) Arrowhead indicates rootlet-like structures in the FLP PCMC. (**K**) Longitudinal ET view of a FLP dendritic branch shows an iteratively bulbous dendrite (arrows) that includes 2–3 MTs, which are tightly packed in the ~60 nm constrictions between bulbs (inset). Scale bars: 100 nm.

## Concluding remarks

The morphological and ultrastructural analyses presented here build on previous work (*Ward et al., 1975*; *Ware et al., 1975*; *Perkins et al., 1986*) to provide a complete 3D description of the anterior sensory anatomy of *C. elegans* at high resolution. Together, these analyses reveal several novel features of *C. elegans* cilia in detail that will inform investigations into neuronal functions. For instance, the IL1/2, OLL/Q, CEP and FLP mechanosensory neurons contain diverse ciliary specializations including the presence of electron-dense disks, deposits of electron-dense material that can be MT-associated, tubular bodies, and flap-like elaborations (as in FLP) at the distal ends, spurs and strings that connect the cilium to the cuticle and surrounding cells, unique axoneme MT organization, distribution and protein association, and the presence of striated or non-striated rootlets and rootlet-like structures. In contrast, the polymodal ASH amphid neurons that also respond to mechanical stimuli (*Kaplan and Horvitz, 1993*; *Hilliard et al., 2005*) contain a prototypical axoneme. This diversity in sensory structures may reflect the distinct types and intensities of mechanical forces transduced by different mechanosensory neurons and/or different transduction mechanisms. As an example, we speculate that the lack of specialized cilia structures in ASH may imply a mode of force transduction that does not involve extensive ciliary deflection perhaps due to the tight arrangement of cilia in the amphid channel pore.

Our analyses also reveal the ultrastructural basis of the intricate branching patterns present at the sensory endings of many head neurons. The specific contributions of these complex structures to neuronal functions remain to be determined. It can be argued that a larger ciliary surface area enhances sensitivity via increased localization of signal transduction molecules (analogous to the situation in photoreceptor outer segments). However, it is currently unclear whether the complex branching pattern in AWA cilia, the larger wing-like structure in AWC, the many microvilli found in AFD, and the two cilia found per dendrite in ADL, ADF, and AWB, all serve to simply increase surface area, or whether the precise topological organization of signaling molecules within these structures is in addition important for signal transduction. In future, further advances in cilia imaging techniques (e.g., *Sahl and Moerner, 2013*; *Su et al., 2013*; *Ye et al., 2013*) and examination of ciliary protein trafficking pathways in these unique cilia types may allow us to describe not only how these structures are formed, but how their morphologies shape, and are in turn shaped, by neuronal functions.

## Materials and methods

### Strains and growth conditions

*C. elegans* animals were maintained at 20°C on standard nematode growth media plates seeded with *E. coli* OP50 bacteria. Wild-type *C. elegans* (Bristol strain N2) was obtained from the *Caenorhabditis* Genetics Center.

### Specimen preparation

Worm samples were prepared by transferring 1-day-old adult hermaphrodite worms to 20% bovine serum albumin (BSA) in M9 buffer in the cavity of an aluminum planchette (type 'A' hat; 100 μm deep, Wohlwend, Switzerland) for high-pressure freezing (*McDonald et al., 2007*). In some cases, animals were treated with 10 mM levamisole and embedded in 2.4% low-melting point agarose pads (*Kolotuev et al., 2009*). We noted that levamisole treatment and agarose embedding resulted in shortened channel cilia lacking distal segments; thus, this method was not used to model the distal ends of amphid channel cilia. The flat surface of another planchette (type 'B' hat) was placed on top to enclose the worms in the planchette's cavity. The specimen–planchette sandwich was rapidly frozen using a Leica EM HPM100 high-pressure freezing system (Leica Microsystems, Vienna, Austria). Freeze-substitution was performed at low temperature (−90°C) over 3 days in a solution containing 1% osmium tetroxide (19,100, EMS),

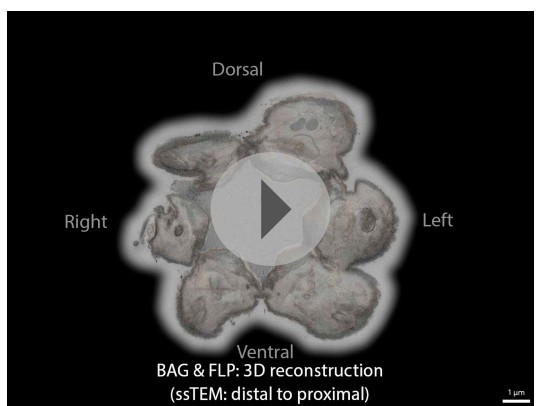

**Video 7**. 3D reconstruction model of BAG and FLP cilia associated with the ILso process.

0.5% glutaraldehyde (16,530, EMS) and 2% water in anhydrous acetone (AC32680-1000, Fisher) using a Leica EM AFS2 freeze-substitution system. The temperature was progressively increased up to 0°C (5°C/hr), and finally brought to 4°C and maintained at this temperature for 1 hr. Samples were washed four times with anhydrous acetone (30 min each), then infiltrated and embedded in Araldite 502/Embed-812 Resin (Araldite—10,900, EMS, Embed-812—14,900, EMS, DDSA—13,710, EMS) at room temperature and polymerized in an oven at 60°C for 3–4 days.

## Serial section TEM

Resin blocks with specimens were in most cases trimmed so that the block face was perpendicular to the longitudinal axis of the worm nose for serial cross-sections, while keeping a small amount of resin around the specimen. For longitudinal images of the nose sensilla, the block face was aligned parallel to the longitudinal axis of the worm nose. Serial ultrathin sections (70-nm thickness) were collected on slot grids covered with Formvar support film, and post-stained with 2% uranyl acetate (0379, Polysciences, Inc., Warrington, PA) for 30 min, and Reynold's lead citrate (Lead nitrate—17,900, EMS, and Sodium citrate—S-279, Fisher) for 15 min. Serial sections were imaged on a Tecnai F20 (200 keV) or F30 (300 keV) transmission electron microscope (FEI, Hillsboro, OR) and recorded using a 2K × 2K charged-coupled device (CCD) camera at 14,500X magnification on the F20 (1.25 nm pixel size), or at 23,000X magnification on the F30 (1 nm pixel size). For large overviews of the cross and longitudinal sections, we acquired montages of overlapping high-magnification images in an automated fashion using the microscope control software SerialEM (*Mastronarde, 2005*).

## Serial section ET

After analyzing TEM images of the serial sections, grids, and sections with structures of interest were prepared for ssET by coating both sides of the serial sections on Formvar with 10-nm colloidal gold fiducials (Sigma-Aldrich, St. Louis, MO) that were previously incubated for 30 min in 5% BSA (SC-2323, Santa Cruz Biotechnology, Inc.) (*Iancu et al., 2006*). Dual-axis tilt series, that is, two orthogonal tilt series of regions of interest, were acquired by tilting the sample from −60 to +60 with 1° increments using the microscope control software SerialEM (*Mastronarde, 2005*) on a Tecnai F20 (200 keV). All images were digitally recorded on a 2K × 2K CCD camera, at 14,500X or 19,000X magnification, resulting in a pixel size of 1.25 nm or 1.04 nm, respectively.

## Image processing

### ssTEM

Image processing to generate and analyze ssTEM 3D reconstructions was automated using manual correction only as needed. Blendmont, a utility of the IMOD software package (*Kremer et al., 1996*) was used to finely align and stitch image tiles into a single montage image for each whole worm cross section. Images were converted from MRC to TIFF format with mrc2tif (IMOD) and then to PNG format (ImageJ) while preserving image resolution. The ir-tools (http://www.sci.utah.edu/download/ncrtoolset) were modified and used for automated section-to-section image registration (*Tasdizen et al., 2010*). Nornir Build Manager software (http://github.com/nornir) was used to coordinate execution of the ir-tools, including the calculation of transforms for each section and the generation of the final volume with registered images (*Anderson et al., 2009*). Briefly, volume assembly began with automatic image registration of all adjacent section pairs. The centermost section was not warped and designated as reference in the volume center. The remaining sections were mapped in relation to the center by serially applying the intervening adjacent section-to-section transforms and warped to correct for distortions resulting from the sectioning process or section folds. Where automatic registration failed between adjacent sections, the alignment was done interactively using Pyre (Nornir) or ir-tweak (ir-tools). The registered serial section images were

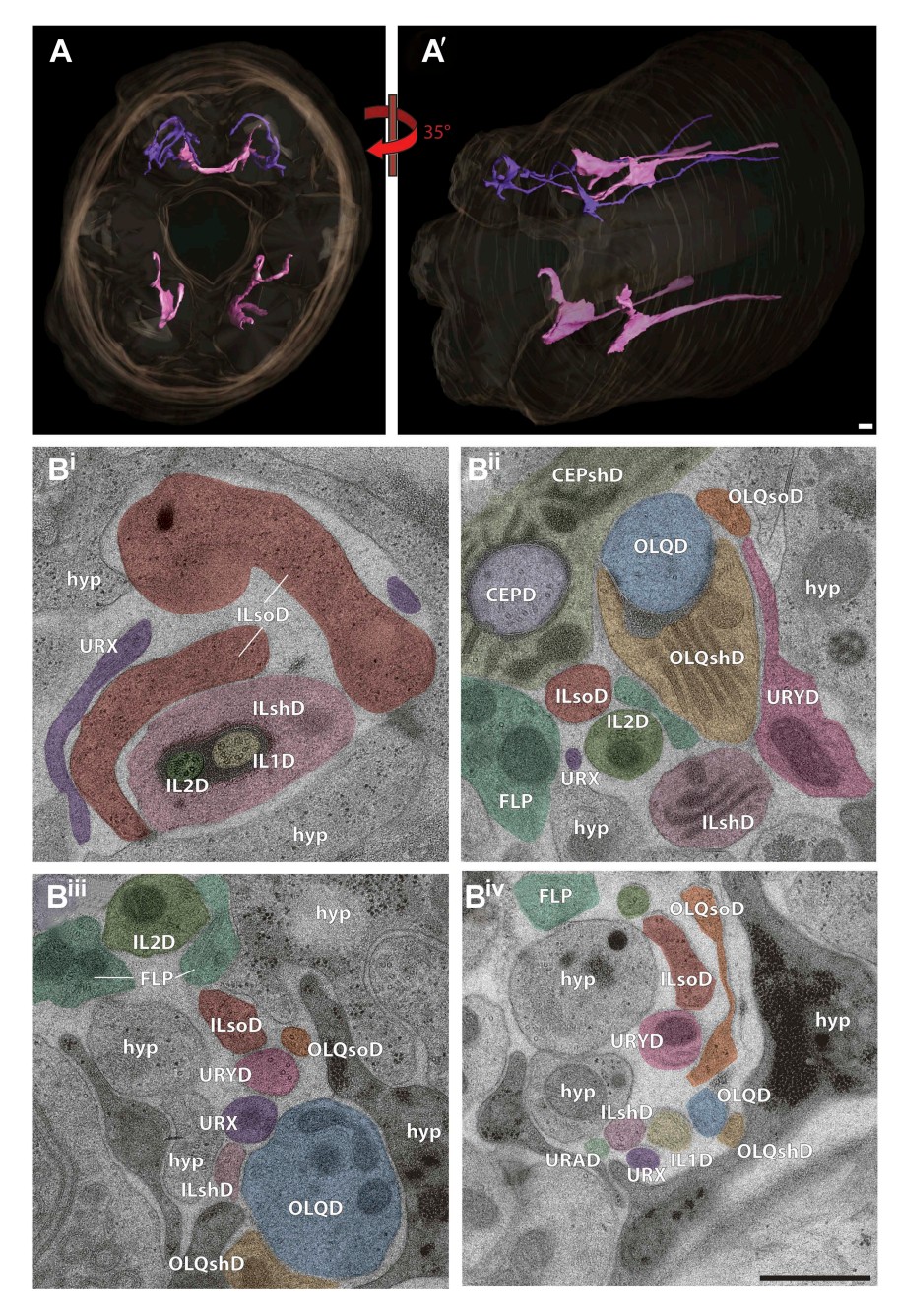

**Figure 16**. Non-ciliated endings of URX and URY. (**A** and **A'**) 3D reconstruction of two dorsal URX (purple) and four dorsal and ventral URY (pink) dendritic endings. (**B^i–iv**) Selected cross-sections from distal (**B^i**) to proximal (**B^iv**). (**B^i**) The tip of URX (purple) terminates around the ILD sensillum with branches around the process of ILsoD. (**B^ii**) URY is found apposed to processes of OLQshD and ILshD. (**B^iii**) URX and URY endings are found near the OLQ dendrite and also surrounded by processes of OLQsoD, ILsoD, and ILshD. Note singlet MTs in URY. (**B^iv**) URX and URY are found sub-dorsally with IL1D, OLQshD, ILshD, OLQsoD, and ILsoD processes as well as hypodermal cells as neighbors. Scale bars: 500 nm.

then converted from PNG format to TIFF (ImageJ) and then to MRC format (tif2mrc/IMOD) to allow stacking of the aligned images into the ssTEM 3D reconstruction (newstack/IMOD) (*Kremer et al., 1996*). Contours of cell membranes and other features were manually traced in IMOD to generate 3D graphical models (3dmod, imodmesh/IMOD) (*Kremer et al., 1996*).

## ssET

3D tomographic reconstructions (dual axis) were generated using etomo from the IMOD software package (*Kremer et al., 1996*; *Mastronarde, 1997*); gold fiducial markers were used for alignment of the tilt series images and the tomograms were calculated in silico using weighted back projection. Serial section tomograms were then aligned, joined, and modeled in 3D using IMOD tools.

## Acknowledgements

We are grateful to Chen Xu for providing training and for management of the Brandeis EM facility, David Hall for extensive discussion and advice, and David Mastronarde for developing IMOD tools. We thank Cori Bargmann, Oliver Blacque, David Hall, Oliver Hobert, Eve Marder, and members of the Sengupta and Nicastro lab for advice and critical comments on the manuscript.

## Additional information

### Funding

| Funder | Grant reference number | Author |
| --- | --- | --- |
| National Institutes of Health | T32 NS007292 | David B Doroquez |
| National Science Foundation | MRI 0722582 | Piali Sengupta, Daniela Nicastro |
| PKD Foundation | 186F09b | David B Doroquez |
| National Institutes of Health | R37 GM56223 | Piali Sengupta |
| National Institutes of Health | P30 NS045713 | Cristina Berciu |
| National Science Foundation | DMR-MRSEC 0820492 | Cristina Berciu |
| National Institutes of Health | R01 EY02576 | James R Anderson |

The funders had no role in study design, data collection and interpretation, or the decision to submit the work for publication.

### Author contributions

DBD, Acquisition of data, Analysis and interpretation of data, Drafting or revising the article; CB, Acquisition of data, Analysis and interpretation of data; JRA, Analysis and interpretation of data, Contributed unpublished essential data or reagents; PS, DN, Conception and design, Analysis and interpretation of data, Drafting or revising the article

## Additional files

### Supplementary files

• Supplementary file 1. (**A**) isMTs in amphid neuron cilia. (**B**) Location of vesicles in sensory cilia.

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
