## [Decision Letter]

Thank you for sending your work entitled “A high-resolution morphological and ultrastructural map of anterior sensory cilia and glia in *C. elegans*” for consideration at *eLife*. Your article has been evaluated by a Senior editor and 4 reviewers, one of whom is a member of our Board of Reviewing Editors. The Reviewing editor and the other reviewers discussed their comments before we reached a decision, and the Reviewing editor has assembled the following comments to help you prepare a revised submission.

All reviewers appreciated the technical quality and potential impact of the work. The one point that the reviewers thought should be strengthened in a revised version of the manuscript is the observation of vesicles in ciliary axonemes (Figure 4—figure supplement 1). This is potentially a very significant new finding that contradicts a dogma in ciliogenesis research. This finding would need better corroboration: is there convincing EM to show the phospholipid bilayer using standard TEM methods in the axonemes – are they unilamellar or multilamellar? Some statistical analysis would be useful e.g., how many vesicles per axoneme, how many per µm of axoneme length etc. Are there crosslinks between the vesicles and the axonemal MTs. These data should then be more prominently presented in the paper.

A revised version also has to contain a significant rewording of the description of previous data by Evans et al., concerning the morphology of MTs in the AWC cilia. You state “Although it was reported that AWC cilia lack sMTs in their distal regions (26), we observed sMTs and dMTs as well as isMTs in the distal and MTs in the far-distal regions of the AWC wing (Figure 10)”. This appears to be a serious misrepresentation of Evans et al., who reported that “Significantly, moving towards the distal tip of the cilium both the singlet and doublet MTs terminated at the same point, approximately 0.5µm below the overlying ciliary membrane”, which is in complete agreement with the current paper. Evans et al. did report that they saw no distal singlet extensions of the doublets, which is also not challenged by the current work.

---

## [Author Response]

*All reviewers appreciated the technical quality and potential impact of the work. The one point that the reviewers thought should be strengthened in a revised version of the manuscript is the observation of vesicles in ciliary axonemes (*Figure 4—figure supplement 1*). This is potentially a very significant new finding that contradicts a dogma in ciliogenesis research. This finding would need better corroboration: is there convincing EM to show the phospholipid bilayer using standard TEM methods in the axonemes – are they unilamellar or multilamellar? Some statistical analysis would be useful e.g., how many vesicles per axoneme, how many per µm of axoneme length etc. Are there crosslinks between the vesicles and the axonemal MTs. These data should then be more prominently presented in the paper*.

To address this issue, we systematically re-examined our EM images of 312 sensory endings from 7 animals. We identified vesicles or vesicle-like structures in the endings of 19 cells of 9 different types including amphid and IL2 neurons. The number of vesicles in each of these sensory endings ranged from 1 to >5. Many of the identified vesicles had clearly defined phospholipid bilayers; none were multilamellar. We confirmed the presence of bilayers by performing ssET on 10 sections containing vesicles.

We also identified the location of these vesicles in each of the sensory endings of the 19 cells. Vesicles were present at the ciliary base in the region where the dMTs flare out in 11 cells, and more distally in the TZ where the dMTs form a tight cylinder in 8 cells. We now show representative ssTEM and ssET images of these vesicles in a new main Figure (Figure 5). At the TZ, vesicles were located between the axoneme and the ciliary membrane in a subset of cilia (called ‘peripheral’ vesicles in new Figure 5), whereas other vesicles were detected within the cylinder of nine outer dMTs (new Figure 5). We also detected vesicles in the distal segment of the axoneme in a single cell (AFD). These are now more clearly shown as a magnified inset in a revised Figure 12. We did not detect crosslinks between the vesicles or between the vesicles and the ciliary membrane, but did observe a possible connection to an axonemal dMT in one case (this can be seen in new Figure 5).

All data described above (identities of cells containing vesicles, vesicle numbers, locations, presence of visible bilayered membrane) are also now included in a new table ([Supplementary-material SD1-data]). These data are also discussed further in the text.

*A revised version also has to contain a significant rewording of the description of previous data by Evans et al., concerning the morphology of MTs in the AWC cilia. You state “Although it was reported that AWC cilia lack sMTs in their distal regions (*[26]*), we observed sMTs and dMTs as well as isMTs in the distal and MTs in the far-distal regions of the AWC wing (*Figure 10*C-E')”. This appears to be a serious misrepresentation of Evans et al., who reported that “Significantly, moving towards the distal tip of the cilium both the singlet and doublet MTs terminated at the same point, approximately 0.5µm below the overlying ciliary membrane”, which is in complete agreement with the current paper. Evans et al. did report that they saw no distal singlet extensions of the doublets, which is also not challenged by the current work*.

We apologize for this error. All references to findings from Evans et al have been corrected.